# Comparisons of Different Methods to Determine Starting Altitudes for Dry Air Atmosphere by GNSS-RO Data

Andrea Andrisani [ID] and Francesco Vespe *[ID]

Agenzia Spaziale Italiana—CGS Matera, 75100 Matera, Italy; andrea.andrisani@asi.it
* Correspondence: francesco.vespe@asi.it

**Abstract:** Boundary profile evaluation (BPV) is an approach proposed in order to estimate water vapor content in the atmosphere. It exploits radio occultation (RO) observations of the signals emitted by the satellites of global navigation systems (GNSS) which are eclipsing (rising) as viewed by a low earth orbit satellite (LEO). BPV requires, as a preliminary step, the estimation of the dry background atmosphere model of refractivity (i.e., obtained from bending angle profiles) to be subtracted from the real observations in order to extract water vapor profiles. The determination of the lowest layer of the atmosphere over which the concentration of water vapor can be deemed negligible is particularly crucial for a correct application of the BPV method. In this study, we have applied three methods to set the starting altitudes for the dry air layers of the atmosphere: (1) by air temperature below some threshold values (for example, 250 K); (2) by "smooth" bending angle profiles in ROs; (3) by saturated water vapor pressure. These methods were tested with thermodynamic and bending angle profiles from 912 radiosonde excursions colocated with RO observations. For every dry air starting altitude we determined the best estimator from each of the three methods. In particular, by comparing those estimators with the quantiles and momenta of the dry air starting altitude distributions, we achieved improvements of up to 50% of the humidity profiles.

**Keywords:** GNSS radio occultation; dry air thresholds; mixing ratio vertical profiles; BPV method

## 1. Introduction

Radio occultation observations continuously measure the deflection angles of an L-band signal (1–2 GHz) from a GNSS satellite to a receiver on a LEO which is viewing the transmitter in the rising−setting phase behind the Earth's disk (see Figure 1). In the RO geometry, the GNSS signal is crossing the atmosphere. Thus, vertical profiles of air refractivity near the Earth's surface can be recovered by these observations because the amosphere interacts with the signal, which modifies the direction (bending angle) and the amplitude. In the last 20 years, thanks to missions such as CHAMP [1], MetOp/GRAS [2] and FORMOSAT/COSMIC-1 and 2 [3], RO observations have provided a huge amount of data to the scientific and user community, suitable for performing global change studies [4] and operational meteorology on a global range [5]. The original ideas using RO for Earth observation were proposed as early as the 1960s [6,7]; the technique was widely applied for studyng Mars' and Venus' atmospheres, with the Mariner and Pioneer space missions [8–10] and the atmosphere of outer planets were probed with the Voyager missions [11]. However, it was only with the launch of the Global Positioning System (GPS) constellation and several LEO satellites that the necessary global coverage of the Earth was achieved, suitable for carrying out an atmospheric survey of the Earth [12]. We want to point out that RO has been applied also between LEO−LEO satellites [13,14], with signals operating on bands other than the L-band of the GNSS.

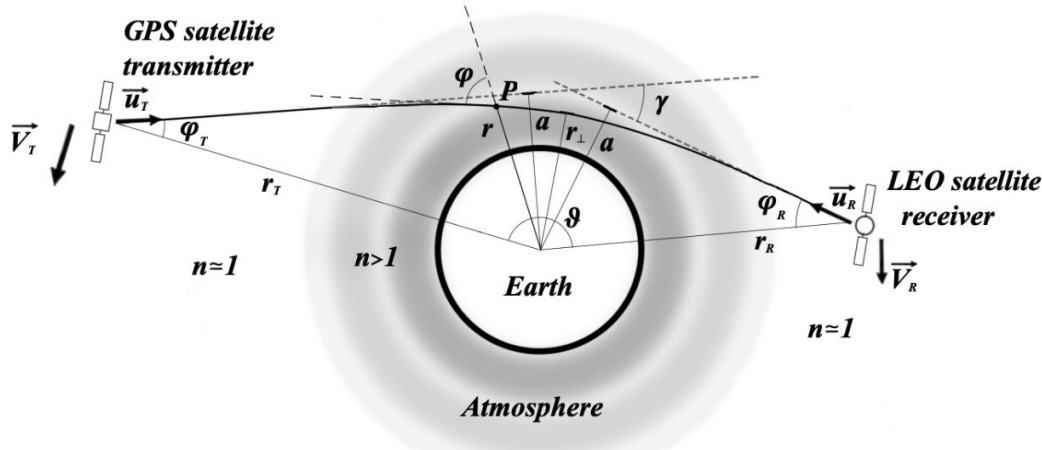

**Figure 1.** Graphical representation of an RO observation. The main RO parameters are also reported. Radio signals from the GPS transmitter deviate from the straight line by a bending angle $\gamma$ towards the LEO receiver, due to the refraction of the atmosphere. Impact parameter of the incoming signal is denoted by $a$, while $\varphi$ is the incident angle of the ray line on the spherical surface of constant refraction index $n$. Bending angles $\gamma$ and impact parameters $a$ are estimated by Doppler shift measurements of the carrier frequency, if the satellites' positions and velocities, respectively $\vec{r}_T$, $\vec{r}_R$ and $\vec{V}_T$, $\vec{V}_R$ are known. More information about the mentioned parameters can be found in [15].

In the absence of water vapor, solely by refractivity, we can estimate the vertical profiles of temperature and pressure via the state equations of gases, the hydrostatic equilibrium and empirical formulas for air refractivity in the L-band, as the Smith and Weintraub microwave refractivity [16] or more recent analogous expressions [17–19]. However, in the presence of water vapor such estimations are no more accurate. Indeed, the temperature derived from ROs by neglecting water vapor concentrations, the so-called dry temperature $T_{dry}$, can be lower than the real temperature $T$ even by several degrees. In contrast, the analogously defined dry pressure $P_{dry}$ usually overestimates the real pressure $P$, with mean deviations that are less pronounced in percentage with respect to those relative to the temperatures. In Section 2, we see that this is a consequence of the high microwave refractivity coefficient of water vapor due to its electric dipole moment.

So, water vapor concentrations cannot be neglected for an accurate determination of the atmospheric temperature and pressure by RO. Besides, water vapor concentration is a fundamental parameter to know in its own right. This "rank deficiency" problem is usually solved in two ways: (1) by applying 1DVAR (NDVAR) techniques or (2) by merging observations with models in a "simple" or "direct" fashion, if we follow Kursinski's classification [20]. As far as 1DVAR or NDVAR techniques are concerned, they consist of an optimization procedure that minimizes the sum of the quadratic errors from two data sources in competition, respectively background data and experimental data, whose relative contributions are weighted by the inverse of the related covariance error matrices [21,22]. The cost function to minimize is usually given in this form:

$$J(x) = \frac{1}{2}\left(y - H(x)\right)^T (E + F)^{-1}(y - H(x)) + \frac{1}{2}(x - x_B)^T B^{-1}(x - x_B), \qquad (1)$$

where $y$ stays for the observed data, $x$ for the data to be recovered, related to the observed data by the forward model $H$, $x_B$ represents the background information, while $B$, $E$ and $F$ are the covariance error matrices, respectively, for the background data, the observed data and the forward model. In the case of ROs, $y$ may represent atmospheric refractivity or rawer experimental data, such as bending angles (BA) or excess Doppler shifts [23–26], $x_B$ is usually given by atmospheric models [27] or by forecasting [5], while, as far as the error covariance matrices estimations are concerned, there is nowadays a wide literature [28–31].

On the other hand, simple models recover humidity concentration profiles by determining dry refractivity first and then operate a simple subtraction from the total refractivity (refractivity due to dry air plus water vapor molecules). Dry refractivity is usually estimated through external information, as temperature profiles [32,33], or by dry refractivity models to be fitted with the RO data in the dry part of the atmosphere, that is, the stratosphere and the upper troposphere. In the last case, the fitting result is successively extrapolated toward the lower troposphere. This procedure is adopted, for example, by the boundary profile evaluation (BPV) method [34,35], a standalone method which applies a constrained data fitting in order to avoid an unphysical negative value for water vapor concentrations [36–38]. Obviously, particular attention must be put on the determination of the starting altitude of the dry part of the atmosphere and its estimation from the RO data: a too high threshold altitude for dry air would affect the accuracy of the extrapolation procedure, while a too low one would falsify the fitting procedure, part of the data to be fitted actually being total refractivity instead of dry refractivity. For example, in [34–38], concerning the BPV method, the Kursinski's criterium [39] was adopted, that is, air at temperatures below 250 K is considered dry. Nevertheless, in the comparative studies [37,38] it was questioned if a different, hopefully more accurate, definition/estimation for the dry air starting altitudes could improve the BPV thermodynamic estimations.

In this work, we firstly analyzed the possible criteria for dry atmospheric conditions. Since water vapor molecules are, more or less, always present in the atmosphere, we have to set a threshold value for humidity concentrations below which air can be safely assumed dry. Obviously, working in the context of RO measurements, such threshold values have to be set according to RO observation accuracies, even if we cannot exclude them, they could find application in other contexts as well. We do this in two ways, depending on whether we measure air humidity by water vapor mixing ratio or by wet refractivity (for a definition of these quantities, see Equations (3) and (10) in Sections 2 and 3, respectively): (1) we define as thresholds those values of water vapor mixing ratios corresponding to significative mean deviations of air dry temperatures and/or pressures from the real ones, where the term significative is related to some statistical considerations; (2) we define as thresholds those values of wet refractivity having the same order of magnitude of the RO dry refractivity uncertainties. Both in (1) and (2) we actually considered a *set* of possible threshold values, rather than a single one: we do this because, depending on the particular method one chooses to investigate the thermodynamic properties of the atmosphere, or depending on the site location, different dry air criteria could be most suited for different users. Finally, the dry air starting altitudes are defined as the highest altitudes corresponding to those where these threshold values are exceeded.

Second, we tried to estimate the previously defined dry air starting altitudes by quantities that can be recovered by RO observations. We considered three methods for this task: (1) by (dry) air temperatures exceeding some threshold values (for example, 250 K, as in [39,40]); (2) by the appearance of irregular patterns in the bending angle vertical profiles, as suggested in [41]; (3) by estimating water vapor mixing ratios and wet refractivities via saturated water vapor pressure, this last derived for example by the Murphy−Koop formula [42]. Such hypotheses were tested with a statistical study concerning 912 thermodynamic and bending angle vertical profiles of the atmosphere, with the first ones derived from experimental radiosonde data colocated with RO events.

The article is structured as follows. In Section 2 we determine more accurately the effects of water vapor over the discrepancy between dry temperature and pressure with respect to the real ones, so as to furnish in Section 3 the threshold values of water vapor concentrations calibrated to the RO response to the thermodynamic of the atmosphere; the analogous values of wet refractivity, calibrated to the dry refractivity uncertainty in RO observations, are given too. In Section 4 we describe the RO derived quantities used to estimate the threshold altitudes as defined in Section 3, whose performances are tested in Section 5. A brief analysis of water vapor concentrations in the stratosphere is given in Section 6, while in Section 7 we draw our conclusions.

## 2. Dry Temperature and Pressure Behaviors Due to Water Vapor Concentrations

In Figure 2, the vertical profiles of dry temperature and pressure, obtained from an RO event that took place at Gresik Island, Indonesia, on 1 January 2009, are reported together with temperatures, pressures and water vapor mixing ratios measured by a balloon excursion. As we can see, temperature−dry temperature differences rise up to 50 degrees in concomitance with increasing mixing ratios, while at the same time the pressure−dry pressure difference decreases up to −140 hPa. Such behavior of dry temperature under-estimation coupled with dry pressure over-estimation is not casual: it is a consequence of the high microwave refractivity coefficient $a_3 \sim 1.73 \times 10^8$ m$^3$·K/kg of the water vapor, due to its electric dipole moment, in the Smith and Weintraub equation:

$$N \cong a_1 \rho_d + \left( a_2 + \frac{a_3}{T} \right) \rho_w, \tag{2}$$

where $\rho_d$, $\rho_w$ are dry air and water vapor concentrations [kg/m$^3$], $a_1 \sim 2.23 \times 10^4$ m$^3$/kg and $a_2 \sim 3.29 \times 10^4$ m$^3$/kg [25] are the induced dipole moment refractivity coefficients for dry air and for water vapor, respectively [15,43]. Let us see this in more detail. Equation (2) can be written as

$$N = a_1 \rho_{dry},$$

where $\rho_{dry}$ is the air density obtained from the measured refractivity by assuming dry atmosphere

$$\rho_{dry} \approx \rho_d \left[ 1 + \left( \frac{a_2}{a_1} + \frac{a_3}{a_1 T} \right) \frac{\rho_w}{\rho_d} \right] = \rho_d \left[ 1 + \left( \frac{a_2}{a_1} + \frac{a_3}{a_1 T} \right) \varrho_r \right],$$

with

$$\varrho_r = \frac{\rho_w}{\rho_d} . \tag{3}$$

the water vapor mixing (density) ratio. For a layer of air subject to a pressure $P$ due to the above air column weighing on it, we get

$$P = \left( \frac{\rho_d}{m_d} + \frac{\rho_w}{m_w} \right) RT \approx \frac{\rho_{dry}}{m_d} RT_{dry}^{(0)}, \tag{4}$$

from the ideal gas state equation, where $m_d = 28.96$ g/mol and $m_w = 18.02$ g/mol are the molar mass of dry air and water vapor, respectively, while

$$T_{dry}^{(0)} \approx T \left[ \frac{1 + \frac{m_d}{m_w} \varrho_r}{1 + \left( \frac{a_2}{a_1} + \frac{a_3}{a_1 T} \right) \varrho_r} \right]. \tag{5}$$

In the typical range of air temperatures, $a_2/a_1 + a_3/(a_1 T) \sim 20 > 1.6 \simeq m_d/m_w$, so $T_{dry}^{(0)}$ is always lower than $T$. Actually, differences between temperatures and dry temperatures are partially mitigated with respect to what Equation (5) suggests, since, in order to determine $T_{dry}$, in (4) we have to replace the real pressures $P$ with $P_{dry}$, the air pressures measured from RO observations by neglecting again water vapor. By assuming for simplicity a constant value for the gravity acceleration $g$, and denoted with $\rho = \rho_d + \rho_w$ the total density, we have:

$$P_{dry}(h) - P(h) = g \int_h^{h_{top}} \left[ \rho_{dry}(z) - \rho(z) \right] dz = g \int_h^{h_{top}} \left[ \frac{a_2}{a_1} + \frac{a_3}{a_1 T(z)} - 1 \right] \rho_w(z) \, dz \geq 0. \tag{6}$$

That is, $P_{dry} \geq P$, at least if no other biases are present in the estimation of $P_{dry}$. Then we have

$$T_{dry} = m_d \frac{P_{dry} - P}{\rho_{dry} R} + T_{dry}^{(0)} \geq T_{dry}^{(0)}. \tag{7}$$

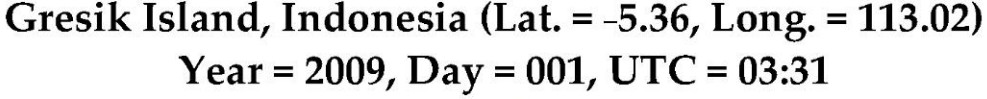

Figure 2 top and bottom panels.

**Figure 2.** Vertical profiles of real temperature $T$ and dry temperature $T_{dry}$ (**top**), and real pressure $P$ and dry pressure $P_{dry}$ (**bottom**), together with mixing ratio $\varrho_r$ for a RO taking place at Gresik Island, Indonesia, on 1 January 2009.

Despite this last correction, $T_{dry}$ can be lower by several degrees than $T$ for nonnegligible values of the mixing ratio, while differences between $P_{dry}$ and $P$ are relatively less pronounced. Temperature and pressure trends, shown in Figure 2, as a particular case, find a more general and accurate description in Figure 3, where the nean deviations (MD) and the root mean square deviations (RMSD) of the dry temperatures and pressures, derived from the 912 RO observations considered in this study, are reported for various altitudes, together with mean values of mixing ratios. Deviations of the dry temperatures

and pressures are calculated with respect to the temperatures and pressures measured by the colocated balloon excursions, that in the following will be considered as the real ones. For pressures, whose values in a vertical profile change up to three orders of magnitude, relative deviations are actually considered. In formula:

$$T_{dry}\ \text{MD} = \sum_{i=1}^{n} \frac{T_i - T_{dry,i}}{n}, \quad T_{dry}\ \text{RMSD} = \sqrt{\sum_{i=1}^{n} \frac{\left(T_i - T_{dry,i}\right)^2}{n}},$$

$$P_{dry}\ \text{MD} = \sum_{i=1}^{n} \frac{P_i - P_{dry,i}}{n \cdot P_i}, \quad P_{dry}\ \text{RMSD} = \sqrt{\sum_{i=1}^{n} \frac{\left(P_i - P_{dry,i}\right)^2}{n \cdot P_i^2}}, \tag{8}$$

where $n$ from now on will denote the number of samples. Temperature and pressure corresponding to given altitudes are obtained via linear interpolation for both the RO and balloon profiles. RO observations come from FORMOSAT-3/COSMIC-1 mission [44], they occurred in 2009 (see Figure 4 for location and date of the events), with the colocated balloon data [45] from the RAwinsonde OBservation (RAOB) program [46]. Colocated events were selected by requiring the following conditions: (1) RO profiles with impact parameters falling below an altitude of 10 km from the Earth's surface; (2) radiosonde excursions reaching altitudes above 21 km; (3) radiosonde excursions without data lacking in their profiles.

From Figure 3 we observe that the mean values of the water vapor mixing ratios $\overline{\varrho_r}$ showed an exponential trend varying with altitude, with non-negligible mean values starting from 10–11 km that increased up to ~$5 \times 10^{-3}$ toward the surface. As far as the temperatures were concerned, almost no biases between real and dry temperature were present at altitudes above ~11 km, while below such values, when water vapor concentrations became non-negligible, MD (and RMSD) monotonically increased up to ~15 K (22 K). For pressures, things were more complicated, since we registered positive biases between data from radiosondes and data from ROs at high altitudes (30 km). Such biases, having different sources than the water vapor, which is absent at those altitudes, showed a decreasing trend even if with an irregular pattern. The water vapor effects on this trend seemed to start at ~8 km, when the pendency of the pattern further decreased (that is, MDs decreased more rapidly below 8 km); besides, the RMSDs inverted the trend and started to increase for altitudes below ~7 km.

In Table 1, we report the dry temperature MDs and dry pressure relative MDs, this time relative to the altitudes at which the given values of the mixing ratio, reported in the first column, were achieved for the first time from the top of the atmosphere. Mean altitudes and mean real temperatures and pressures are reported too, together with the number of samples, which varied since in some vertical profiles the higher values of the mixing ratio were never achieved. The balloon data was linearly interpolated in order to match the altitudes of the (denser) RO vertical profiles.

**Table 1.** Mean values of altitude, temperature, temperature−dry temperature and relative pressure−dry pressure differences together with the number of samples corresponding to given values of mixing ratio. Uncertainties are also given in brackets.

| $\varrho_r$ | $n$ | $\overline{h}$ (km) | $\overline{T}$ (K) | $\overline{P}$ (hPa) | $\overline{T - T_{dry}}$ (K) | $\overline{\left(P - P_{dry}\right)/P}$ (%) |
|---|---|---|---|---|---|---|
| $1 \times 10^{-6}$ | 912 | 14.49 (±2.94) | 217.12 (±9.32) | 146.92 (±84.52) | 0.14 (±2.17) | 0.49 (±1.16) |
| $1 \times 10^{-5}$ | 911 | 11.52 (±3.24) | 220.51 (±7.92) | 229.87 (±98.76) | 0.16 (±2.33) | 0.48 (±1.18) |
| $2 \times 10^{-5}$ | 911 | 10.28 (±3.06) | 224.24 (±7.52) | 273.16 (±100.86) | 0.21 (±2.20) | 0.44 (±1.15) |
| $3 \times 10^{-5}$ | 910 | 9.55 (±2.95) | 227.10 (±7.82) | 302.27 (±104.00) | 0.29 (±2.18) | 0.40 (±1.11) |
| $4 \times 10^{-5}$ | 910 | 8.92 (±2.74) | 229.41 (±7.84) | 327.78 (±103.96) | 0.44 (±2.31) | 0.37 (±1.08) |
| $5 \times 10^{-5}$ | 910 | 8.43 (±2.56) | 231.46 (±8.13) | 349.06 (±102.65) | 0.62 (±2.38) | 0.35 (±1.07) |

**Table 1.** *Cont.*

| $\varrho_r$ | $n$ | $\overline{h}$ (km) | $\overline{T}$ (K) | $\overline{P}$ (hPa) | $\overline{T - T_{dry}}$ (K) | $\overline{\left(P - P_{dry}\right)/P}$ (%) |
|---|---|---|---|---|---|---|
| $7.5 \times 10^{-5}$ | 910 | 7.72 ($\pm$2.37) | 235.14 ($\pm$8.83) | 383.67 ($\pm$105.51) | 0.93 ($\pm$2.49) | 0.31 ($\pm$1.01) |
| $1 \times 10^{-4}$ | 909 | 7.22 ($\pm$2.17) | 237.86 ($\pm$8.88) | 409.62 ($\pm$106.76) | 1.19 ($\pm$2.57) | 0.29 ($\pm$0.97) |
| $2.5 \times 10^{-4}$ | 896 | 5.78 ($\pm$2.14) | 247.70 ($\pm$9.21) | 500.86 ($\pm$130.56) | 2.59 ($\pm$3.51) | 0.10 ($\pm$0.72) |
| $5 \times 10^{-4}$ | 850 | 4.70 ($\pm$2.17) | 255.27 ($\pm$9.04) | 580.10 ($\pm$148.72) | 4.59 ($\pm$4.79) | −0.15 ($\pm$0.64) |
| $7.5 \times 10^{-4}$ | 798 | 4.13 ($\pm$2.13) | 259.39 ($\pm$8.25) | 625.53 ($\pm$154.79) | 6.01 ($\pm$5.34) | −0.32 ($\pm$0.63) |
| $1 \times 10^{-3}$ | 726 | 3.88 ($\pm$2.05) | 262.15 ($\pm$7.92) | 645.69 ($\pm$152.67) | 7.46 ($\pm$5.92) | −0.45 ($\pm$0.65) |

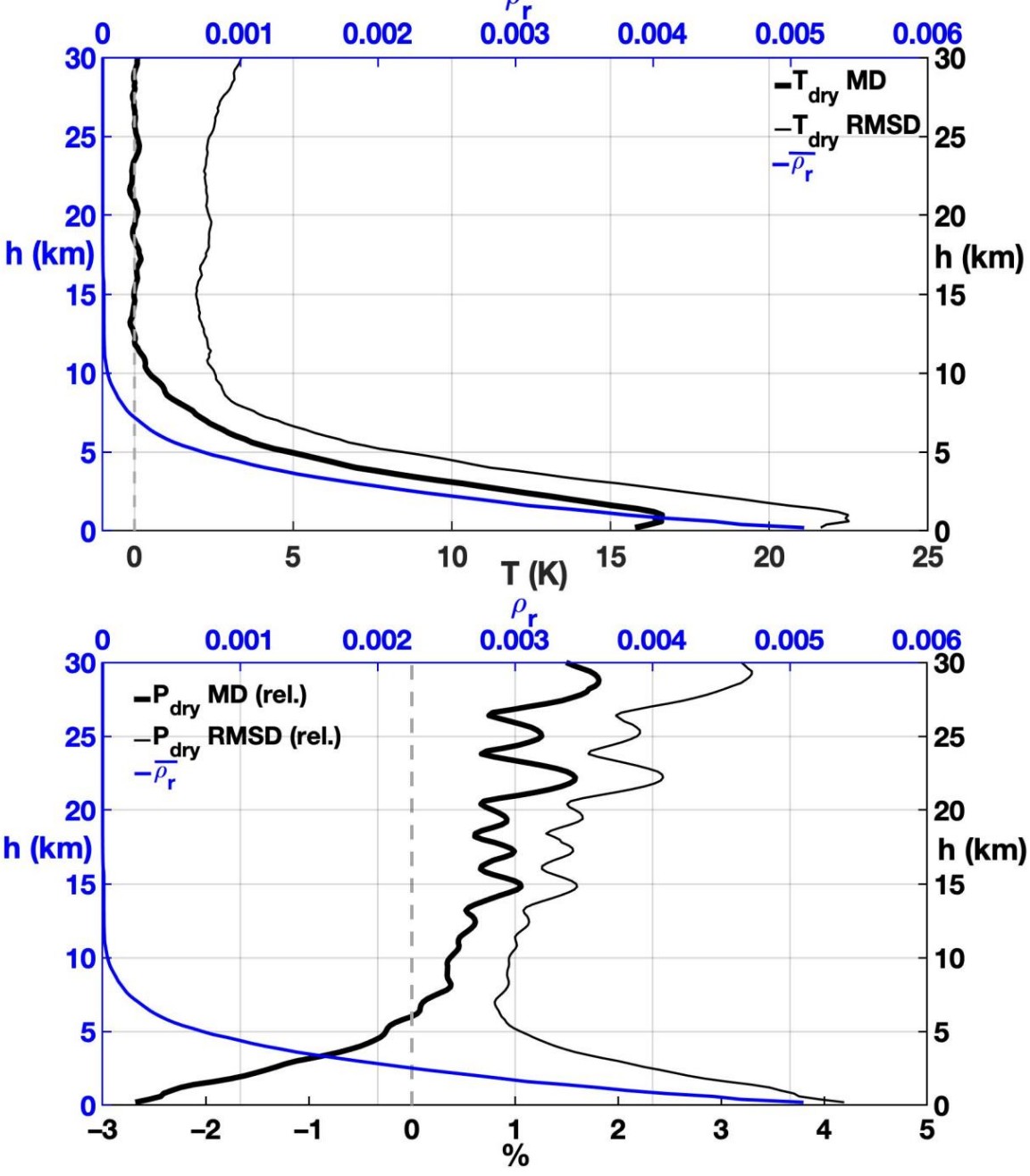

**Figure 3.** (**Top**) Vertical profiles for the mixing ratio mean values (blue) and for the dry temperature mean deviations (thick black line) and the root mean square deviations (thin black line) with respect to the air temperatures measured by balloon excursions (**bottom**). Analogous profiles for pressures.

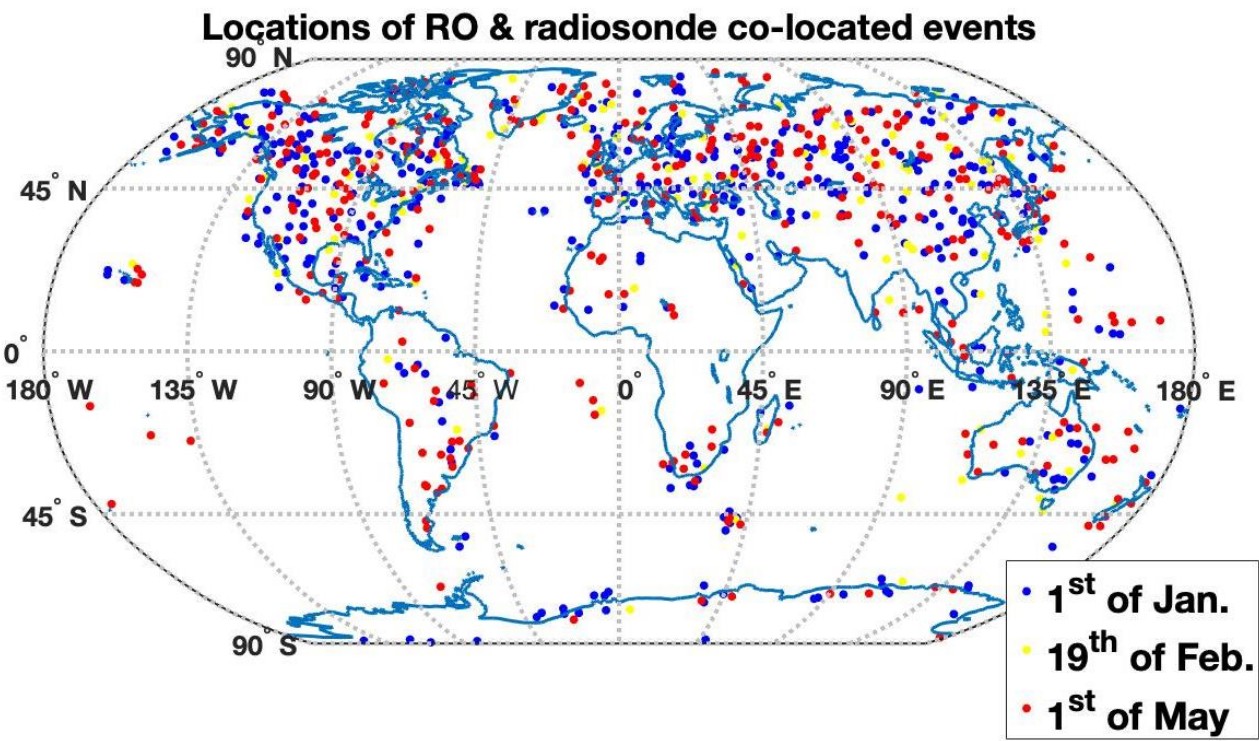

**Figure 4.** Event locations for the RO observations colocated with radiosonde excursions considered in this study, taking place on the 1st of January (422 events), the 19th of February (118 events) and the 1st of May (372 events).

## 3. Dry Air Starting Altitudes Definitions

### 3.1. Definitions by Water Vapor Mixing Ratios

From Figure 3 we deduce that water vapor firstly affects dry temperature profiles, while the effects on dry pressures are visible only at lower altitudes. So, for a definition of dry air starting altitudes by mixing ratios, calibrated to the RO observations, we can limit ourselves to the dry temperature deviations alone.

By statistical analysis of the 912 vertical profiles considered in this study, the differences between the balloon temperatures and dry temperatures $\Delta T = T - T_{dry}$ result in a mean value of 0.0251 K and a standard deviation of 2.4974 K, in the altitude range from 20 km to 30 km where water vapor is negligible. Assuming an identical normal distribution for $\Delta T$ even in the range 0–20 km in the absence of water vapor, from a unilateral mean test we obtain the mean value $\overline{\Delta T} = 0.16$ K, relative to the subset of altitudes with a mixing ratio of $10^{-5}$ (see Table 1), is a significative increment of the mean, with a level of confidence of almost 95% (for $\overline{\Delta T} = 0.21$ K, corresponding to a mixing ratio of $2 \times 10^{-5}$, we have an increment of the mean with a level of confidence exceeding 99%). That is, dry temperature deviations that we register when $\varrho_r = 10^{-5}$ cannot be attributed to the typical dispersions of temperature in dry air, but are actually the effect of this (low) value of water vapor concentration. Such deviations increase with the increasing of the mixing ratio, and at $\varrho_r \sim 2.5 \times 10^{-4}$, corresponding to a temperature close to the Kursinski's temperature threshold of 250 K, we register a deviation that exceeds the standard deviation of $\Delta T$ itself.

From these considerations we define the dry air starting altitude $h_{\varrho, r_i}$ by means of the following threshold values for the mixing ratio:

$$h_{\varrho, r_i} \equiv \text{altitude at which } \varrho_r \gtrsim r_i \text{ for the first time from the top of the atmosphere toward the ground} \qquad (9)$$

$$\left( r_i = 10^{-5}, 5 \times 10^{-5}, 10^{-4}, 1.5 \times 10^{-4}, 2 \times 10^{-4}, 2.5 \times 10^{-4} \right).$$

The chosen values for $r_i$ are those of relevant interest from the data exposed in Table 1 and from the previous statistical analysis, together with some intermediate values. We remark that in the following we will not try to determine which values of $r_i$ are, in general,

the best suited, since this is a consideration that actually depends on the particular method one choses to integrate RO observations in order to take account of the water vapor. We will just see how good the estimations of these starting altitudes are, by the quantities that we will define in Section 4. Nevertheless, we have to observe that for $r_i = 2.5 \times 10^{-4}$, corresponding to Kursinski's temperature of 250 K, we register quite a large deviation of dry temperature from the real one (~2.5 K).

### 3.2. Definitions by Wet Refractivity

Another possible criterion to establish a dry air starting altitude concerns the wet refractivity, in particular when compared with the uncertainty in refractivity in an RO observation due the remaining sources of error. We say that air ceases to be dry when the wet refractivity, calculated from the known values of water vapor concentrations, exceed the uncertainty in the refractivity estimation from RO observations, so becoming the main source of error for $N$.

Wet refractivity $N_w$ is proportional to water vapor concentration $\rho_w$ via the second and the third terms in the Smith and Weintraub relation (1)

$$N_w \cong \left(a_2 + \frac{a_3}{T}\right)\rho_w, \tag{10}$$

Given $\delta_{N_d}$ the uncertainty in dry refractivity, we set

$$h_N \equiv \text{altitude at which } N_w \gtrsim \delta_{N_d} \text{ for the first time from the top of the atmosphere toward the ground.} \tag{11}$$

So, at $h < h_N$, wet refractivity can be seen as the main source of error for dry refractivity. Starting from this altitude, wet refractivity has to be estimated.

About $\delta_{N_d}$, we considered two estimations:

1. $\delta_{N_d} = 0.05$
2. $\dfrac{\delta_{N_d}}{N_d} = \begin{cases} 0.002 & if \quad 10 \text{ km} \geq h > 20 \text{ km} \\ 10^{-0.07 \cdot h - 2} & if \qquad h < 10 \text{ km} \end{cases}$ ,

where $N_d$ is the dry refractivity. The first is an absolute estimation and comes directly from the FORMOSAT-3/COSMIC-1 data, which include for each RO a vertical profile for the uncertainty in dry refractivity. After considering all the vertical profiles in our study, taking for each profile the values of $\delta_{N_d}$ in the altitude range from 7 to 13 km—where we expect to encounter water vapor for the first time—and after mediating all of these values, we obtain the number 0.05. Refractivity uncertainties at these altitudes, as reported in COSMIC-1 data, are estimated from Doppler frequency uncertainties through error propagation [47], so they just take into account the so-called measurement or structural errors, such as instrument thermal noise, reference link noise and atmospheric scintillations [48,49] while neglecting the modeling errors concerning, in particular, the local spherical symmetry assumption.

The second one is a relative estimation and comes from a schematized version of the relative uncertainty profile reported in Kursinski et al. [15]. Contrary to the previous estimations, they considered the deviations from local spherical symmetry as well as the source of errors and uncertainty increments below 10 km, which, as we can see in Figure 5 where expression 2 is graphically reported, were mainly due to the possible presence of horizontal gradients in the atmosphere.

We then define two threshold altitudes, $h_{N1}$ and $h_{N2}$, respectively, if we consider expression 1 or 2 for $\delta_{N_d}$. In general, the second estimation for refractivity uncertainty being greater than the first, we will usually have $h_{N1} > h_{N2}$.

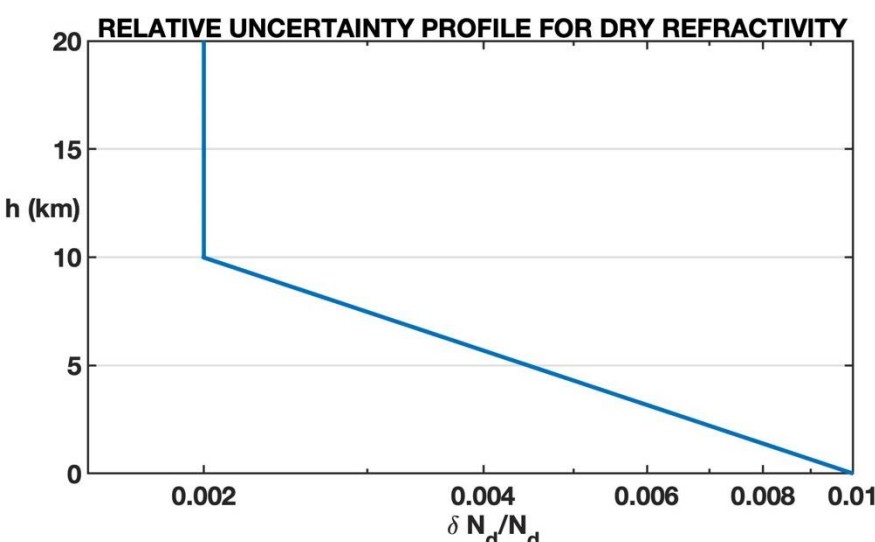

**Figure 5.** Graphical representation of the RO uncertainty profile 2, deduced from Figure 13 of [15].

## 4. RO Estimators for Dry Air Starting Altitudes

In this Section, we consider three atmospheric quantities, derivable by RO data, which can be used to estimate the dry air starting altitudes defined in Section 3. They are the (dry) air temperature, the difference of the bending angle from its total variation and the saturated water vapor pressure.

### 4.1. Air Temperature

From Table 1, we see that the given values of mixing ratio correspond to air temperature values with small relative uncertainties (3–4%), if compared to the uncertainty concerning pressure, approximatively equal to 25–55%, or concerning altitude, equal to 20–50%. So, we reverse the scheme of Table 1 and estimate mixing ratios starting from (dry) air temperatures.

In Kursinski [39], an air temperature of 250 K or less in the troposphere is considered a safe condition for negligible amounts of water vapor. Such an empirical hypothesis is motivated by some thermodynamic considerations, water vapor not having enough cinematic energy to break the chemical bonds in the water vapor droplets at those temperatures. In Figure 6, we report a vertical profile of the mixing ratio together with the profile of dry temperatures for an RO event colocated with a radiosonde excursion: as one can see, an appreciable amount of water vapor is concomitant with a dry air temperature exceeding 250 K.

In our study we tested Kursinski's hypothesis, together with additional threshold temperatures. In particular, we will compare the previously defined altitudes $h_{\varrho,r_i}$, $h_{N1}$ and $h_{N2}$ of Equations (4) and (5), respectively, with

$$h_{T_i} \equiv \text{altitude at which the dry temperature } T_{dry} \geq T_i \text{ for the first time from the tropopause toward the ground.} \quad (12)$$

$$(T_i = 210\,\text{K}, \ 215\,\text{K}, \ 220\,\text{K}, \ 225\,\text{K}, \ 230\,\text{K}, \ 235\,\text{K}, \ 240\,\text{K}, \ 245\,\text{K}, \ 250\,\text{K}, \ 255\,\text{K})$$

Contrary to definitions (9) and (11), $h_{T_i}$ is calculated starting from the tropopause. Such a request is motivated by the temperature gradient inversion at the tropopause. Indeed, in the stratosphere we encounter favorable thermodynamic conditions for water vapor formation. Nevertheless, before they could reach the stratosphere from the troposphere, water vapor molecules cross the colder tropopause, where most of them usually go in condensation, that is, in a hydrostatically unfavorably denser state for the higher layers of the atmosphere. A thermodynamic based estimator such as (12), concerning air temperature, would often result in false positives by localizing water vapor in the stratosphere. So, we

have to limit its application to the troposphere, de facto renouncing the few occasions on which water vapor could nevertheless be detected in the stratosphere (see Section 6 for more details).

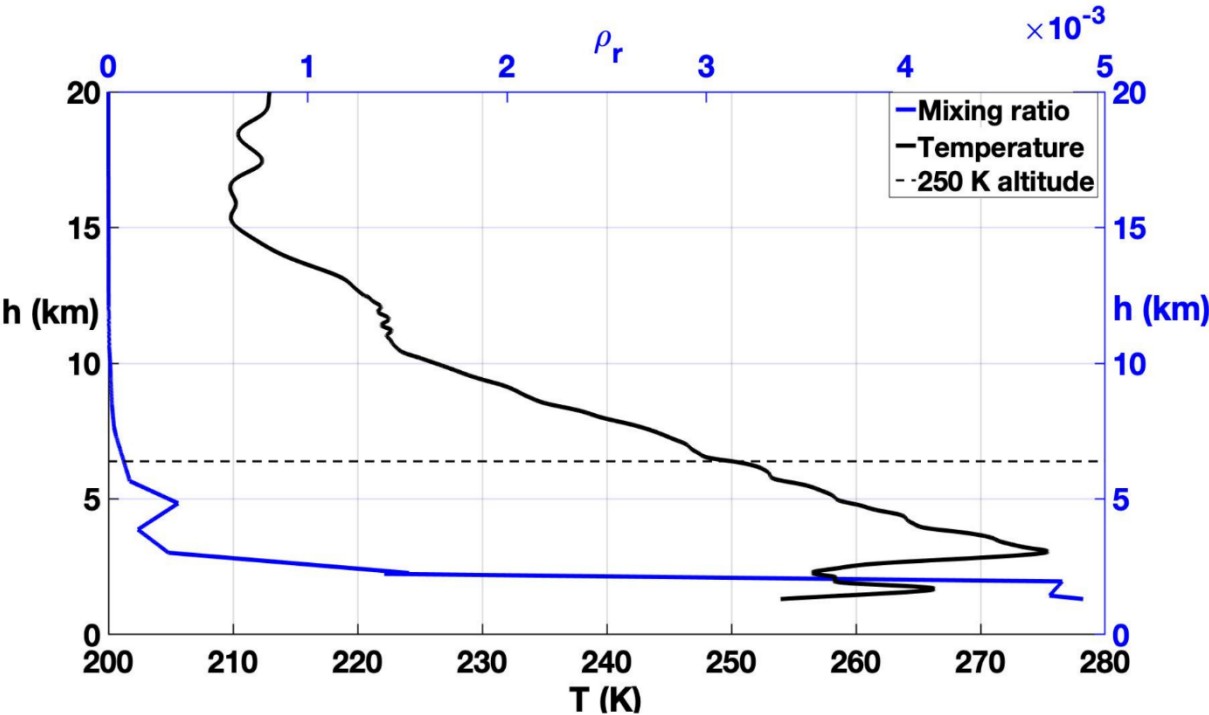

**Figure 6.** Mixing ratio profile (blue) and temperature profile (black) from an RO event colocated with a radiosonde excursion. The 250 K threshold altitude for air temperature is also plotted.

Observe that when tropospheric temperatures are greater than $T_i$, then, according to definition (12), $h_{T_i} = h_{trop}$, with $h_{trop}$ the tropopause altitude. Concerning the tropopause definition used in this study, we refer to Section 6 as well.

Finally, we test Kursinski's and similar hypotheses for both the dry temperatures, recovered by RO observations, and the real temperatures, from balloon data. We will denote with $h_{T_i}^{(RO)}$ and $h_{T_i}^{(b)}$ the altitudes relative to the first and the second case, respectively.

### 4.2. Difference of the Bending Angle from Its Total Variation

Rao et al. [41] noted that the bending angle profiles usually show an irregular or a "random walk" pattern in correspondence of the tropopause and of water vapor as well (see Figure 7). While in [41] an algorithm was set for a "radio tropopause altitude" definition, no analogous rule was given concerning water vapor appearance.

We try to estimate threshold altitudes for water vapor concentrations directly from bending angle profiles $\gamma(h)$ by looking at their total variation [50], that we remember was defined as:

$$TV_\gamma(h) = \gamma(h_0) + \int_h^{h_0} \left| \frac{d\gamma}{dh}(z) \right| dz,$$

where $h_0$ is a reference altitude to be defined, while $h \leq h_0$. For its ability to detect sharp variations of a function, the total variation operator is often used in denoising problems [51], in deconvolution problems [52], in segmentation [53], and in inpainting [54,55].

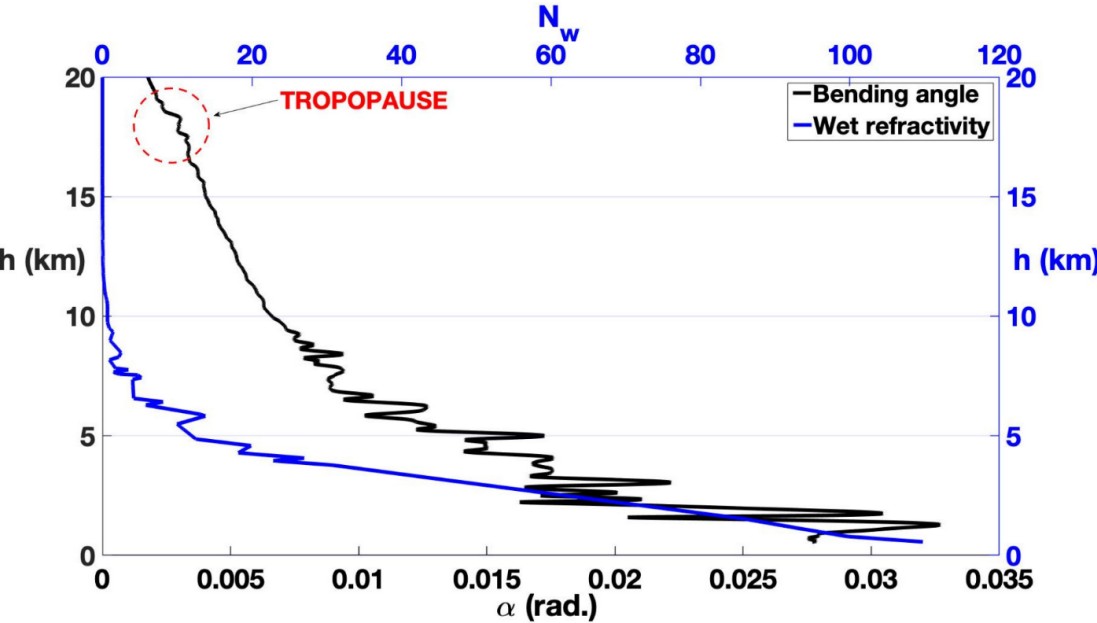

**Figure 7.** Bending angle (blue) and wet refractivity (black) profiles from an RO event colocated with a radiosonde atmospheric excursion. A slight irregular behavior for the bending angle at ~18 km is in correspondence with the tropopause altitude (red circle).

In general, $TV_\gamma$ deviates from $\gamma$ when $\gamma$ shows non-monotone behavior: then a significative deviation of $TV_\gamma$ from $\gamma$ could indicate water vapor presence, according to [41] (see Figure 8).

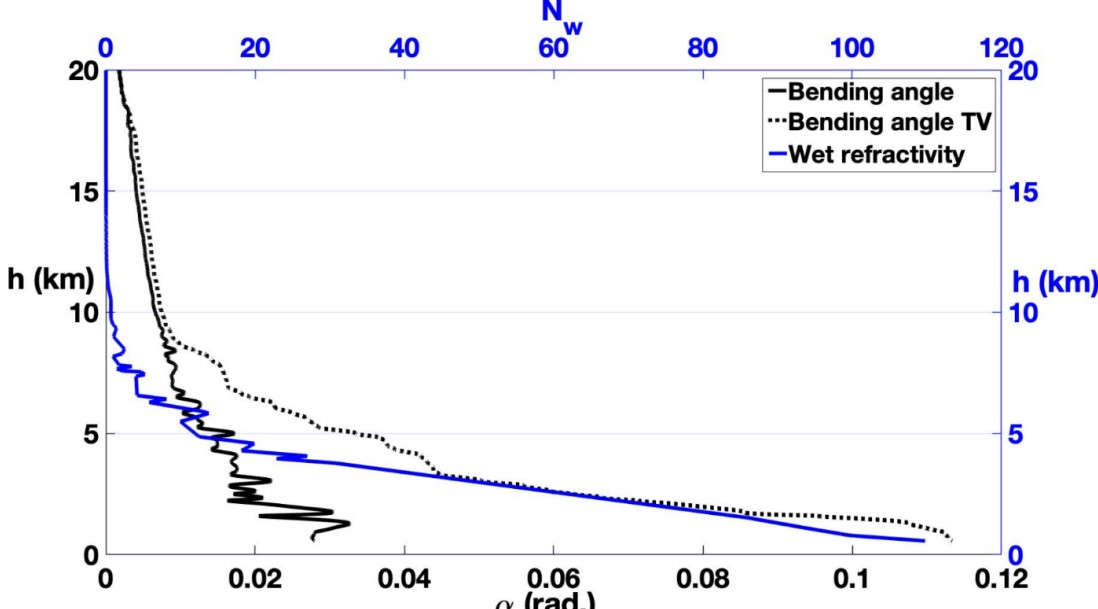

**Figure 8.** Bending angle $\gamma$ (black), total variation of bending angle $TV_\gamma$ (dotted black) and wet refractivity $N_w$ (blue) profiles from the same event in Figure 6. Total variation deviates from $\alpha$ in correspondence with an increase in wet refractivity.

Defined $\widetilde{\gamma} = TV_\gamma - \gamma$, we considered the following altitudes to estimate the dry air starting altitudes of Section 2:

$$h_{\widetilde{\gamma}_i} \equiv \text{altitude at which } \widetilde{\gamma} \gtrsim \widetilde{\gamma}_i \text{ for the first time from } h_0 \text{ toward the ground.} \qquad (13)$$

$$\left(\widetilde{\gamma}_i = 5 \times 10^{-6} \text{ rad}, 10^{-5} \text{ rad}, 5 \times 10^{-5} \text{ rad}, \ 10^{-4} \text{ rad}, 1.5 \times 10^{-4} \text{ rad}, 2 \times 10^{-4} \text{ rad}, 2.5 \times 10^{-4} \text{ rad}\right)$$

We set both $h_0 = 25$ km and $h_0 = h_{trop}$, denoting with $h_{\widetilde{\gamma}_i}^{(25)}$ and $h_{\widetilde{\gamma}_i}^{(tp)}$ the altitude calculated in the first and in the second case, respectively. Observe that in the first case, at least in principle, we can detect water vapor concentration even in the lower stratosphere, contrary to the previous method based on air temperature.

With the $\widetilde{\gamma}_i$ values reported in (13), we saw that we could not properly estimate the dry air starting altitudes corresponding to the higher values of $r_i$ in (9), while we could cover the remaining cases corresponding to the lower/medium ones. On the other hand, considering further, greater values for $\widetilde{\gamma}_i$ with respect to those in (7) would be useless, since the corresponding $h_{\widetilde{\gamma}_i}$ as defined in (13) would present large dispersions, due to the extremely random character of the bending angle in the presence of large amounts of water vapor. For this reason, this method will not be applied to the mixing ratio starting altitudes $h_{\varrho,r_i}$ with $r_i > 10^{-4}$.

### 4.3. Saturated Water Vapor Pressure

Saturated water vapor pressure $P_s$ gives an upper bound for water vapor concentration in the atmosphere, usually being $P_w \leq P_s$ with the exception of a few cases of supersaturation [56]. So, as long as the theoretical value of $P_S$ estimated from RO data does not exceed certain cut-off values, the hypotheses of dry air can be maintained.

An accurate prediction of the saturated water vapor pressure in the atmosphere is not an easy task, given the various elements to take into consideration: water drop radius dimensions/ice crystal structures, supercooled metastable state of water, salt and aerosol concentrations, and so on (see [57,58] for more details). As a preliminary study we will consider the following equation, relating $P_s$ (hPa) with the temperature $T$ (K):

$$P_s(T) = \begin{cases} P_{s;i}(T) & if & T \leq 233.15 \\ P_{s;w}(T) & if & T \geq 253.15 \\ P_{s;i}(T) + \frac{T-233.15}{20}[P_{s;w}(T) - P_{s;i}(T)] & if & 233.15 < T < 253.15 \end{cases}. \qquad (14)$$

Here $P_{s;i}$ and $P_{s;w}$ are, respectively, the saturation pressure for ice and water relative to flat surfaces. Such a choice, similar to that adopted in [59] even if concerning different intervals, is suggested by the homogeneous solidification of water, occurring at $\sim -40\,°C$ [57], and by some studies [60] concerning the heterogeneous ice nucleation by deposition mode, that does not seem to occur at temperatures lower than $\sim -30 \div -20\,°C$. We remember that deposition is the only mode in which water vapor solidifies in the presence of aerosols without a transient state of liquefication (but see Marcolli [61], on ice nucleation in particle capillaries). Concerning $P_{s;i}$ and $P_{s;w}$, there are various definitions in the literature; here we have adopted the equations of Murphy and Koop [42]:

$$\ln P_{w;i} = 9.550426 - \frac{5723.265}{T} + 3.53068 \ln T - 0.00728332 \cdot T - \ln 100,$$

$$\ln P_{w;s} = 54.842763 - \frac{6763.22}{T} - 4.210 \ln T + 0.000367 \cdot T +$$

$$\tanh[0.0415(T - 218.8)]\left(53.878 - \frac{1331.22}{T} - 9.44523 \ln T + 0.014025 \cdot T\right) - \ln 100. \quad (15)$$

In Figure 9, we report the transition curve for water vapor as adopted in this study.

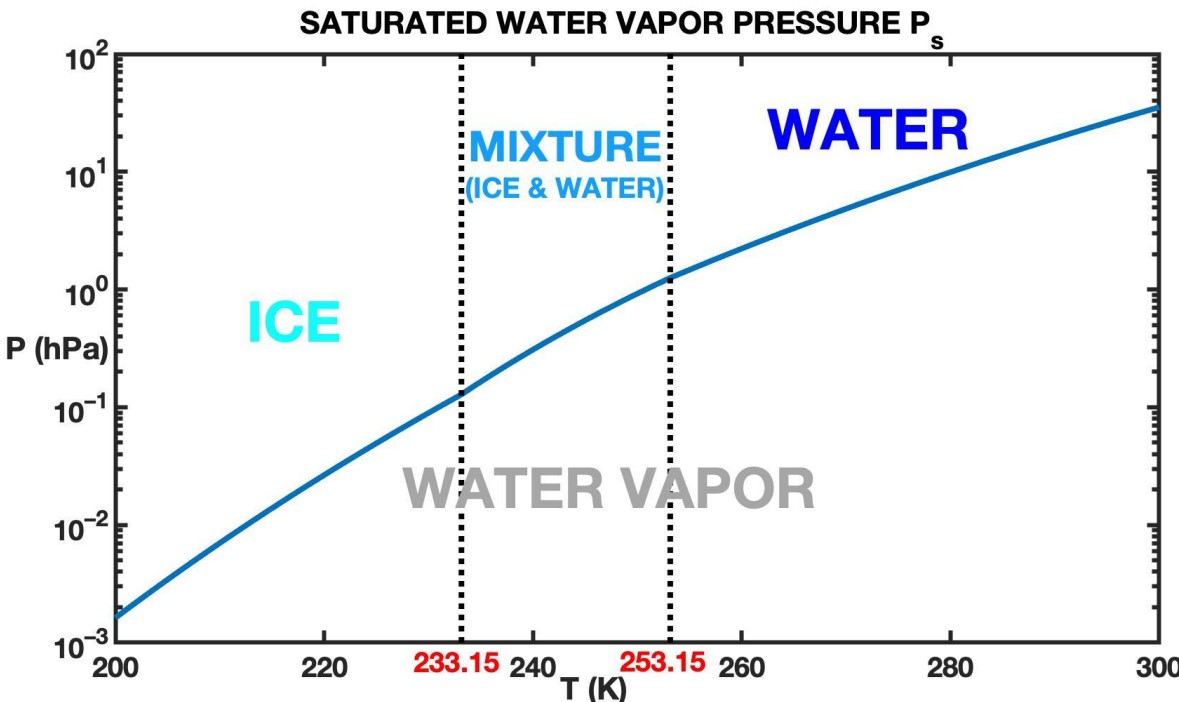

**Figure 9.** Phase transition curve between water vapor and ice and between water vapor and water used in this study. For "mixture" we intend the intermediate region where the saturated water vapor pressure is given by a linear combination of the saturated pressure with respect to ice and water.

With $P_s$ given by the dry temperature profiles calculated from the RO data, we can define

$$P_w^{(s,RH)} = RH \cdot P_s, \qquad (16)$$

with $0 \leq RH \leq 100$ indicating a fixed value of relative humidity. $P_w^{(s,RH)}$ is an estimation of the wet partial pressure. From it and from the dry pressures and temperature, we can easily derive the mixing ratio and the wet refractivity, and consequently the quantities $h_{\rho,r^i}^{(s,RH)}$, $h_{N1}^{(s,RH)}$, $h_{N2}^{(s,RH)}$ in an analogous way to the quantities to compare $h_{\rho,r^i}$, $h_{N1}$, $h_{N2}$ defined in Section 3. However, for the same reasons exposed in Section 4.1 concerning $h_{T_i}$ definition, the quantities $h_{\rho,r^i}^{(s,RH)}$, $h_{N1}^{(s,RH)}$, $h_{N2}^{(s,RH)}$—collectively denoted as $h^{(s,RH)}$—will be calculated starting from the tropopause and not from the top of the atmosphere. For relative humidity, we set $RH = 30\%$, a characteristic value for the tropopause/high troposphere [59,62,63], where we expect to meet threshold conditions for the lower value of humidity concentration, and $RH = 40\%$ for higher values of humidity concentrations, usually occurring at lower quotes.

This method, based on water vapor saturation pressure, is conceptually similar to that based on air temperature exposed in Section 4.1, since the Murphy and Koop equations which we use to calculate $P_s$ depend just on temperature alone. With respect to the air temperature method, now we can obtain quantities that better match the terms that we want to estimate. The downside is that we have to make aprioristic assumptions about the air's relative humidity.

## 5. Results

As said before, for our statistical study we considered 912 RO observations colocated with radiosonde excursions, on the 1st of January, the 19th of February and the 1st of May of 2009. The site locations are reported in Figure 4. As one can see, the events cover all over the world, even if they are mainly concentrated at the northern middle latitudes.

For all these locations we determined the altitudes $h_{T_i}^{(RO)}$, $h_{\widetilde{\gamma}_i}$, $h^{(s,RH)}$ by RO observations and $h_{\varrho,r_i}$ and $h_N$ from radiosonde excursions data. Concerning the former quantities, we calculated their mean deviation (MD), the root mean square deviation (RMSD) and the correlation coefficients with respect to the latter ones. For the dry air starting altitudes $h_{\varrho,10^{-5}}$, $h_{\varrho,5\times10^{-5}}$, $h_{\varrho,10^{-4}}$, $h_{N1}$ and $h_{N2}$, we determined the best estimators among the three families of altitudes $h_{T_i}^{(RO)}$, $h_{\widetilde{\gamma}_i}$, $h^{(s,RH)}$ here considered, by varying $T_i$, $\widetilde{\gamma}_i$ and $RH$, respectively, (for $h_{\widetilde{\gamma}_i}$ we consider both the cases $h_0 = 25$ km and $h_0 = h_{trop}$). For the lower starting altitudes $h_{\varrho,1.5\times10^{-4}}$, $h_{\varrho,2\times10^{-4}}$ and $h_{\varrho,2.5\times10^{-4}}$, concerning the greatest values of mixing ratio here considered, estimations were not performed with the $h_{\widetilde{\gamma}_i}$ altitudes, the selected values of $\widetilde{\gamma}_i$ being too low for a good approximation. The best estimators were selected by requiring that the sum of the RMSD with the absolute value of the MD was minimum (at parity of this sum, those with the highest correlation coefficient): we also calculated the coefficients $a$, $b$ of their linear regressors, with the relative RMSDs.

Nevertheless, for our scope, rather than a linear regressor we needed an optimal overestimator, that is, a new quantity—defined for example by a polynomial function of the estimator—by which we can overestimate the majority of the dry air starting altitudes, minimizing at the same time the MDs. We achieved this with the straight line $h^{(0.9)}$, a first order polynomial of the estimator, whose detailed definition is reported in Section 5.1. So, for each best estimator we calculated the coefficients of the line $h^{(0.9)}$ and the relative MD, too. All these data are reported in Table 2.

### 5.1. Mixing Ratio Starting Altitudes Estimati

In Figure 10, we show the distributions for the dry air starting altitudes defined by the mixing ratio. As one can immediately note, $h_{\varrho,10^{-5}}$, $h_{\varrho,5\times10^{-5}}$ and partly $h_{\varrho,10^{-4}}$ have a bimodal character; we will see below that the second mode is mainly due to the starting altitudes lying in the stratosphere. Concerning the other starting altitudes, they are essentially unimodal and can be well approximated by Gaussians as part of a slightly positive skewness.

In Figures 11 and 12 we show the scatterplots of the altitudes $h_{\varrho,5\times10^{-5}}$ and $h_{\varrho,10^{-4}}$ with their best estimators, respectively, $h_{230\text{ K}}^{(RO)}$, $h_{235\text{ K}}^{(RO)}$ for the dry air temperature altitudes, $h_{3\times10^{-4}rad}^{(tp)}$, $h_{1.5\times10^{-4}rad}^{(tp)}$ for the bending angle-total variation ones, and $h_{\varrho,5\times10^{-5}}^{(s,30)}$, $h_{\varrho,10^{-4}}^{(s,30)}$ for those defined by the saturation pressure of water vapor. In Figure 13, there are the scatterplots of the altitudes $h_{\varrho,2.5\times10^{-4}}$, with the best estimators $h_{245\text{ K}}^{(RO)}$ and $h_{\varrho,2.5\times10^{-4}}^{(s,40)}$. For the other dry air starting altitudes, all the relevant data concerning the estimation process are collected in Table 2.

In Figures 11 and 12, we reported with red points those events for which $h_{\varrho,5\times10^{-5}}$ and $h_{\varrho,10^{-4}}$ fall in the stratospheric region. By projecting those points onto the $y$-axis, we see that the altitudes $h_{\varrho,5\times10^{-5}}$ and $h_{\varrho,10^{-4}}$ higher than 15 km, corresponding in Figure 10 to the position of the second modes, mainly refer to the stratosphere. The same is true for $h_{\varrho,10^{-5}}$. So, we can conclude that the bimodal character of the dry air starting altitude distributions for the lower values of the mixing ratio are due to water vapor concentration in the stratosphere. The percentage of the starting altitudes falling in the stratosphere are reported in Table 2.

**Table 2.** Best estimators of the dry air starting altitudes. Data below give, in row order: (1) mean value and standard deviation for the starting altitude; (2) stratospheric probability for the starting altitude; (3) starting altitude best estimator; (4), (5) and (6) MD, RMSD and correlation coefficient r between the starting altitude and its best estimator; (7) and (8) a and b coefficients of the linear regressor $h_{lin.\ regr.} = a \cdot h_{best\ est.} + b$ ; (9) RMSD of the linear regressor; (10) and (11) a and b coefficients of the altitude $h^{(0.9)} = a \cdot h_{best\ est.} + b.$ ; (12) MD of the $h^{(0.9)}$ altitude. Data obtained by omitting stratospheric altitudes are reported in brackets.

| | | | $\varrho$ | | | | | | N | |
| | | | $h_{\varrho,10^{-5}}$ | $h_{\varrho,5\times10^{-5}}$ | $h_{\varrho,10^{-4}}$ | $h_{\varrho,1.5\times10^{-4}}$ | $h_{\varrho,2\times10^{-4}}$ | $h_{\varrho,2.5\times10^{-4}}$ | $h_{N1}$ | $h_{N2}$ |
|---|---|---|---|---|---|---|---|---|---|---|
| | **Main interval (km)** | | 11.52 ± 3.24 (9.86 ± 2.02) | 8.43 ± 2.56 (8.01 ± 1.90) | 7.22 ± 2.17 (7.13 ± 1.99) | 6.59 ± 2.15 | 6.14 ± 2.14 | 5.78 ± 2.14 | 10.31 ± 2.48 (9.55 ± 1.67) | 7.63 ± 2.98 (7.24 ± 2.42) |
| | **Stratospheric probability (%)** | | 34.76 | 5.37 | 0.99 | 0.44 | 0.11 | 0 | 20.29 | 4.71 |
| **Temperature (RO)** | **Best estimator** | | $h_{210\ \mathbf{K}}^{(RO)}$ (*) | $h_{230\ \mathbf{K}}^{(RO)}$ | $h_{235\ \mathbf{K}}^{(RO)}$ | $h_{240\ \mathbf{K}}^{(RO)}$ | $h_{240\ \mathbf{K}}^{(RO)}$ | $h_{245\ \mathbf{K}}^{(RO)}$ | $h_{215\ \mathbf{K}}^{(RO)}$ (**) | $h_{230\ \mathbf{K}}^{(RO)}$ |
| | MD (km) | | −0.52 (1.66) | −0.25 (0.25) | 0.21 (0.31) | 0.03 | 0.50 | −0.07 | 0.25 (1.37) | −0.18 (1.12) |
| | RMSD (km) | | 3.65 (2.22) | 2.46 (1.31) | 1.71 (1.42) | 1.71 | 1.72 | 1.82 | 2.78 (1.90) | 2.74 (1.97) |
| | r | | 0.08 (0.73) | 0.42 (0.76) | 0.65 (0.74) | 0.65 | 0.67 | 0.61 | 0.16 (0.69) | 0.46 (0.75) |
| | Linear regressor | a | 0.14 (0.83) | 0.59 (0.81) | 0.77 (0.80) | 0.74 | 0.76 | 0.66 | 0.24 (0.68) | 0.76 (1.01) |
| | | b (km) | 10.00 (0.75) | 3.61 (1.36) | 1.46 (1.14) | 1.72 | 1.12 | 2.03 | 7.78 (2.07) | 1.40 (−1.21) |
| | | RMSD (km) | 3.23 (1.39) | 2.33 (1.23) | 1.65 (1.34) | 1.63 | 1.58 | 1.69 | 2.45 (1.21) | 2.64 (1.61) |
| | $h^{(0.9)}$ | a | −0.49 (0.91) | 0.80 (0.92) | 0.91 (0.95) | 0.98 | 1.04 | 0.93 | −0.78 (0.83) | 1.00 (1.11) |
| | | b (km) | 21.69 (0.90) | 2.95 (1.66) | 1.74 (1.41) | 1.59 | 0.81 | 2.35 | 22.91 (1.99) | 0.89 (−0.32) |
| | | MD (km) | 4.75 (1.53) | 1.07 (1.27) | 1.27 (1.33) | 1.48 | 1.58 | 1.86 | 4.32 (1.41) | 1.42 (1.63) |
| **Bending Angle** | **Best estimator** | | $h_{10^{-4}rad}^{(25)}$ | $h_{1.5\times10^{-4}rad}^{(tp)}$ | $h_{3\times10^{-4}rad}^{(tp)}$ | - | - | - | $h_{5\times10^{-6}rad}^{(tp)}$ | $h_{3\times10^{-4}rad}^{(tp)}$ |
| | MD (km) | | −0.10 (1.98) | 0.17 (0.71) | 0.21 (0.31) | — | - | - | −0.05 (0.99) | −0.08 (0.40) |
| | RMSD (km) | | 4.85 (4.07) | 3.73 (3.02) | 3.26 (3.10) | — | - | - | 3.41 (2.49) | 3.94 (3.33) |
| | r | | −0.01 (0.29) | 0.20 (0.40) | 0.31 (0.36) | — | - | - | 0.13 (0.55) | 0.31 (0.33) |
| | Linear regressor | a | −0.01 (0.16) | 0.14 (0.24) | 0.21 (0.23) | - | - | — | 0.12 (0.34) | 0.18 (0.25) |
| | | b (km) | 11.60 (7.92) | 7.26 (5.89) | 5.66 (5.45) | - | - | — | 9.07 (5.95) | 6.30 (5.32) |
| | | RMSD (km) | 3.24 (1.93) | 2.53 (1.74) | 2.06 (1.85) | - | - | — | 2.46 (1.39) | 2.92 (2.28) |
| | $h^{(0.9)}$ | a | −0.13 (0.16) | 0.20 (0.30) | 0.36 (0.36) | - | - | — | −0.30 (0.31) | 0.37 (0.26) |
| | | b (km) | 17.66 (10.54) | 9.11 (7.66) | 7.07 (6.97) | - | - | — | 17.49 (8.06) | 7.54 (8.84) |
| | | MD (km) | 4.62 (2.62) | 2.42 (2.29) | 2.52 (2.53) | - | - | — | 4.11 (1.78) | 3.07 (3.18) |

**Table 2.** *Cont.*

| | | | $\varrho$ | | | | | | N | |
|---|---|---|---|---|---|---|---|---|---|---|
| | | | $h_{\varrho,10^{-5}}$ | $h_{\varrho,5\times10^{-5}}$ | $h_{\varrho,10^{-4}}$ | $h_{\varrho,1.5\times10^{-4}}$ | $h_{\varrho,2\times10^{-4}}$ | $h_{\varrho,2.5\times10^{-4}}$ | $h_{N1}$ | $h_{N2}$ |
| | **Best estimator** | | $h_{\varrho,10^{-5}}^{(s,30)}$ | $h_{\varrho,5\times10^{-5}}^{(s,30)}$ | $h_{\varrho,10^{-4}}^{(s,30)}$ | $h_{\varrho,1.5\times10^{-4}}^{(s,40)}$ | $h_{\varrho,2\times10^{-4}}^{(s,40)}$ | $h_{\varrho,2.5\times10^{-4}}^{(s,40)}$ | $h_{N1}^{(s,30)}$ | $h_{N2}^{(s,40)}$ |
| Saturation | | MD (km) | −0.83 (1.39) | −0.27 (0.24) | −0.14 (−0.04) | 0.29 | 0.20 | 0.13 | −0.12 (0.93) | 0.10 (0.61) |
| | | RMSD (km) | 3.79 (2.09) | 2.61 (1.48) | 1.84 (1.54) | 1.83 | 1.91 | 1.93 | 2.78 (1.63) | 3.11 (2.21) |
| | | r | 0.05 (0.70) | 0.40 (0.74) | 0.65 (0.73) | 0.66 | 0.66 | 0.67 | 0.15 (0.67) | 0.44 (0.69) |
| | Linear regressor | a | 0.08 (0.70) | 0.48 (0.66) | 0.64 (0.66) | 0.63 | 0.58 | 0.57 | 0.22 (0.68) | 0.45 (0.58) |
| | | b (km) | 10.68 (1.94) | 4.51 (2.53) | 2.70 (2.44) | 2.23 | 2.46 | 2.42 | 8.06 (2.39) | 4.14 (2.69) |
| | | RMSD (km) | 3.23 (1.45) | 2.35 (1.27) | 1.66 (1.35) | 1.62 | 1.60 | 1.59 | 2.45 (1.24) | 2.67 (1.74) |
| | $h^{(0.9)}$ | a | −0.54 (0.82) | 0.70 (0.77) | 0.74 (0.74) | 0.81 | 0.77 | 0.73 | −0.64 (0.83) | 0.57 (0.71) |
| | | b (km) | 21.99 (2.30) | 3.85 (2.94) | 3.24 (3.18) | 2.48 | 2.90 | 3.35 | 20.81 (2.31) | 4.92 (3.39) |
| | | MD (km) | 4.69 (1.71) | 1.09 (1.29) | 1.26 (1.33) | 1.46 | 1.67 | 1.91 | 3.98 (1.43) | 1.67 (1.70) |

(*) $h_{210\,K}^{(RO)} = h_{trop}^{(RO)}$ in the 67% of cases. (**) $h_{215\,K}^{(RO)} = h_{trop}^{(RO)}$ in the 41% of cases.

Stratospheric water vapor also has the effect of "ruining" the estimations of $h_{\varrho,r_i}$ by quantities that, with the exception of $h_{\widetilde{\gamma}_i}^{(tp)}$, are not designed to be applied above the tropopause. In Table 2, for those dry air altitudes with non-negligible stratospheric components in their distribution, all the statistical data are recalculated by omitting the stratospheric contribution. These additional data are reported by parenthesis: as one can see, they indicate a consistent improvement in the estimation process. For example, in the estimation of $h_{\varrho,10^{-5}}$, $h_{\varrho,5\times10^{-5}}$ and $h_{\varrho,10^{-4}}$ by the $h_{T_i}^{(RO)}$ and $h_{\varrho,r_i}^{(s,RH)}$ altitudes, we register correlation coefficients of 0.70–0.74 when we limit ourselves to the troposphere, contrary to a correlation coefficient of 0.40–0.42 for $h_{\varrho,5\times10^{-5}}$ or even of 0.05–0.08 for $h_{\varrho,10^{-5}}$ when we consider the stratosphere as well.

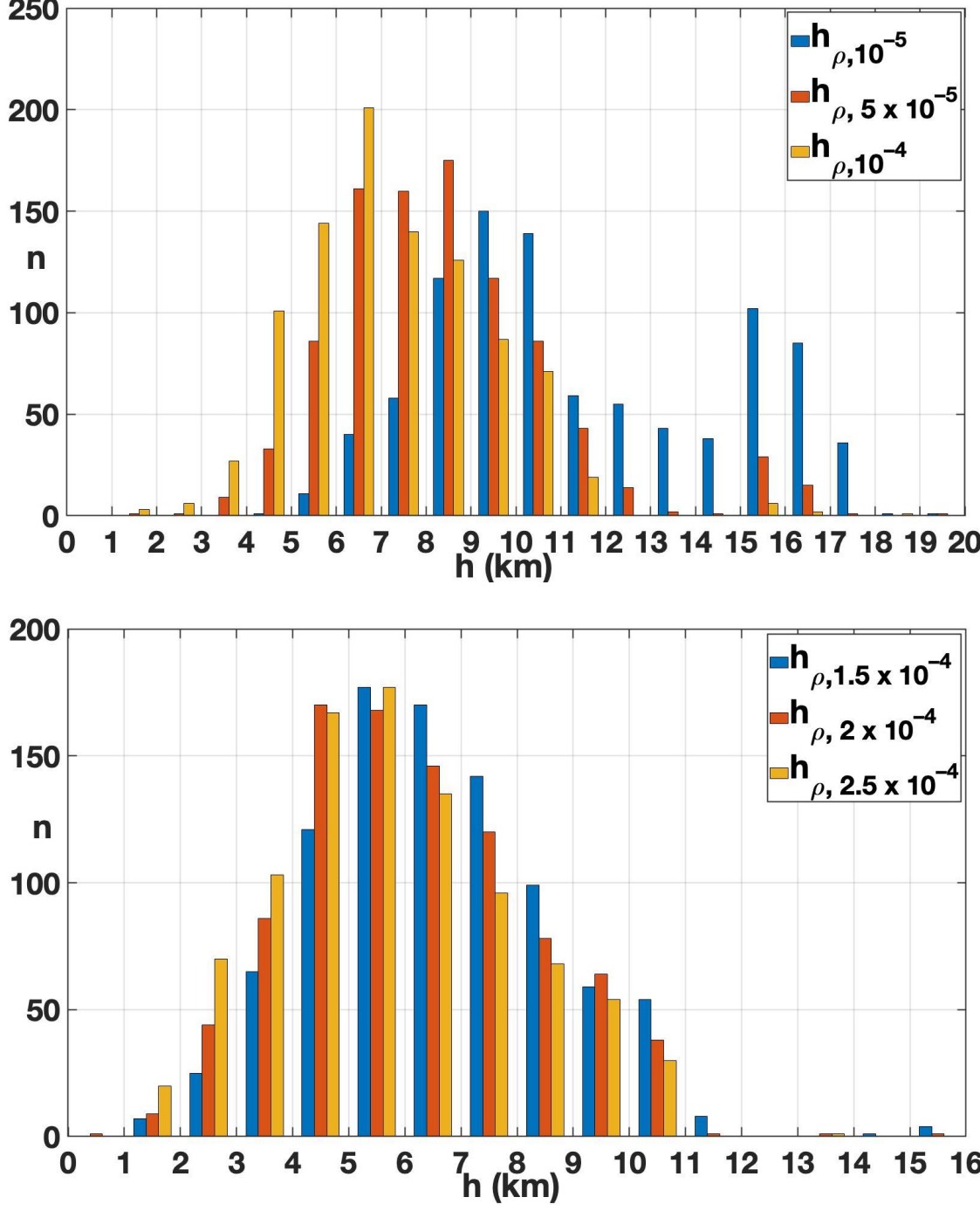

**Figure 10.** Distribution histograms for the dry air starting altitudes $h_{\varrho,10^{-5}}$, $h_{\varrho,5\times10^{-5}}$, $h_{\varrho,10^{-4}}$ (**top**) and $h_{\varrho,1.5\times10^{-4}}$, $h_{\varrho,2\times10^{-5}}$, $h_{\varrho,2.5\times10^{-4}}$ (**bottom**). Means and standard deviations for these distributions are reported in Table 2.

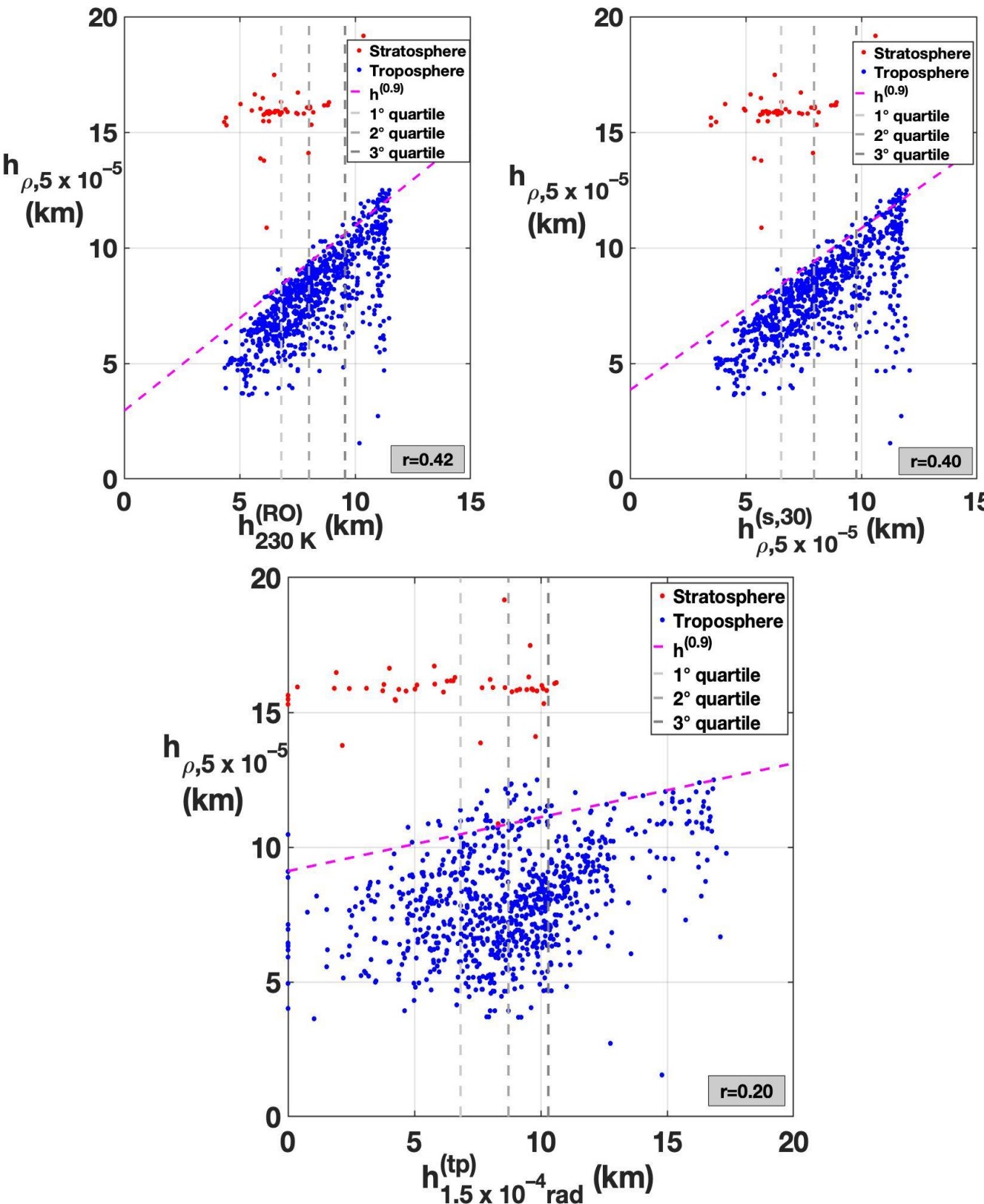

**Figure 11.** Scatterplots among the dry air starting altitudes $h_{\varrho,5\times10^{-5}}$ and the three best estimators $h_{230\,\mathrm{K}}^{(RO)}$ (**top left**), $h_{\varrho,5\times10^{-5}}^{(s,30)}$ (**top right**) and $h_{1.5\times10^{-4}\mathrm{rad}}^{(tp)}$ (**bottom**). Correlation coefficients (r) are reported bottom right, for each graph. Red points denote starting altitude falling in the stratosphere, blue points in the troposphere. Purple broken lines denote the $h^{(0.9)}$ straight lines, while grey broken lines the estimators' quartiles.

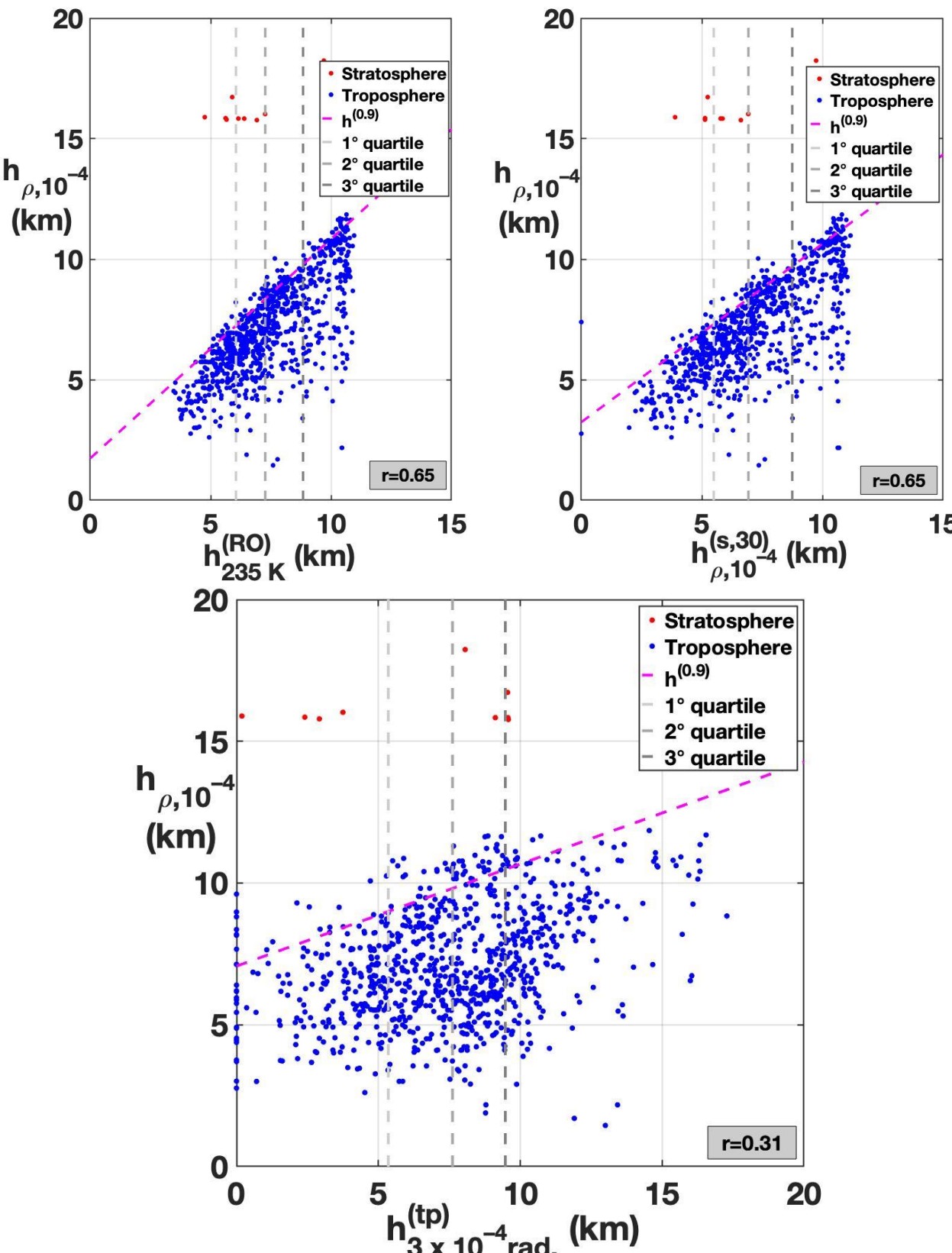

**Figure 12.** Scatterplots among the dry air starting altitudes $h_{\varrho,10^{-5}}$ and the three best estimators $h_{235\,K}^{(RO)}$ (**top left**), $h_{\varrho,10^{-4}}^{(s,30)}$ (**top right**) and $h_{3\times10^{-4}rad}^{(tp)}$ (**bottom**). Correlation coefficients (r) are reported bottom right, for each graph. Red points denote starting altitude falling in the stratosphere, blue points in the troposphere. Purple broken lines denote the $h^{(0.9)}$ straight lines, while grey broken lines the estimators' quartiles.

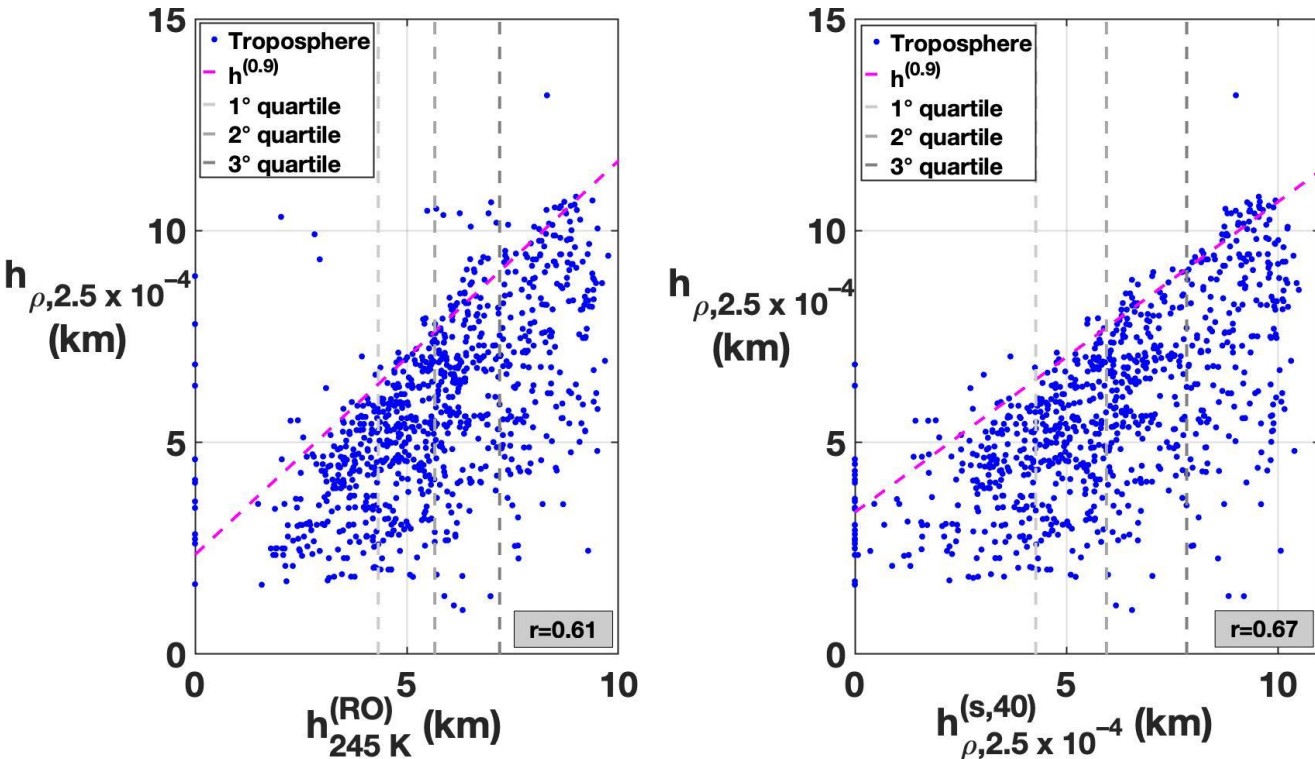

**Figure 13.** Scatterplots among the dry air starting altitudes $h_{\varrho,10^{-5}}$ and the two best estimators $h_{245\,K}^{(RO)}$ (**left**) and $h_{\varrho,10^{-4}}^{(s,30)}$ (**right**). Correlation coefficients (r) are reported bottom right, for each graph. Purple broken lines denote the $h^{(0.9)}$ straight lines, while grey broken lines the estimators' quartiles.

Concerning the other, lower, dry air starting altitudes, with almost no data in the stratosphere, we have correlation coefficients of ~0.65, slightly worse than those relative to the starting altitude defined by lower mixing ratios when the stratospheric data are omitted. This is mainly due to two reasons. First, as we can see from Table 1, starting from $\varrho_r = 3 \times 10^{-5}$, dry temperature deviations from real temperatures increase not only in mean but even in variance. So, we expect a more scattered distribution of $h_{T_i}^{(RO)}$ and $h_{\varrho,r_i}^{(s,RH)}$ for higher values of $T_i$ and $r_i$, than is actually the case if we compare the graphs of Figure 13 with the analogous graphs of Figure 11 or Figure 12. This can also be seen by looking at Table 3, where the same data concerning $h_{T_i}^{(RO)}$ are applied with the altitudes $h_{T_i}^{(b)}$ defined by the real temperatures. For the starting altitudes $h_{\varrho,1.5\times10^{-4}}$, $h_{\varrho,2\times10^{-4}}$ and $h_{\varrho,2.5\times10^{-4}}$, we have correlation coefficients equal to 0.71, 0.72 and 0.74, so not too far from the correlation coefficients concerning the other starting altitudes when the stratospheric samples are neglected. Second, we adopted the convention to set the estimating altitudes equal to zero if, in the RO vertical profiles, the threshold values for the dry air temperatures, for the $\widetilde{\gamma}_i$ and for the "saturated" mixing ratios were never reached, even when the RO data do not cover the first kilometers of the atmosphere near the ground. This rarely happens with the $h_{T_i}^{(RO)}$ and $h_{\varrho,r_i}^{(s,RH)}$ best estimators of $h_{\varrho,r_i}$, if $r_i \leq 10^{-4}$, but it sometimes occurs when $r_i > 10^{-4}$, as the points along the *y*-axis in the scatterplots of Figure 13 testify. Those points also contribute to reducing the correlation coefficients.

**Table 3.** Best estimators of the dry air starting altitudes by the $h_{T_i}^{(b)}$ altitudes, concerning real temperatures. Data below give, in row order: (1) mean value and standard deviation for the starting altitude; (2) stratospheric probability for the starting altitude; (3) starting altitude Best estimator; (4), (5), (6) and (7) MD, RMSD and correlation coefficient r between the starting altitude and its Best estimator; (7) and (8) a and b coefficients of the linear regressor $h_{lin.\ regr.} = a\cdot h_{best\ est.} + b$; (9) RMSD of the linear regressor; (10) and (11) a and b coefficients of the altitude $h^{(0.9)} = a\cdot h_{best\ est.} + b$; (12) MD of the $h^{(0.9)}$ altitude. Data obtained by omitting stratospheric altitudes are reported in brackets.

| | | | $\varrho$ | | | | | | N | |
|---|---|---|---|---|---|---|---|---|---|---|
| | | $h_{\varrho,10^{-5}}$ | $h_{\varrho,5\times10^{-5}}$ | $h_{\varrho,10^{-4}}$ | $h_{\varrho,1.5\times10^{-4}}$ | $h_{\varrho,2\times10^{-4}}$ | $h_{\varrho,2.5\times10^{-4}}$ | $h_{N1}$ | $h_{N2}$ |
| Main interval (km) | | 11.52 ± 3.24 (9.86 ± 2.02) | 8.43 ± 2.56 (8.01 ± 1.90) | 7.20 ± 2.17 (7.13 ± 1.99) | 6.59 ± 2.15 | 6.14 ± 2.14 | 5.76 ± 2.16 | 10.31 ± 2.48 (9.55 ± 1.67) | 7.63 ± 2.98 (7.24 ± 2.42) |
| Stratospheric presence (%) | | 34.76 | 5.37 | 0.99 | 0.44 | 0.11 | 0 | 20.29 | 4.71 |
| **Temperature (b)** Best estimator | | $h_{210\ \mathbf{K}}^{(b)}$ (*) | $h_{230\ \mathbf{K}}^{(b)}$ | $h_{240\ \mathbf{K}}^{(b)}$ | $h_{240\ \mathbf{K}}^{(b)}$ | $h_{245\ \mathbf{K}}^{(b)}$ | $h_{250\ \mathbf{K}}^{(b)}$ | $h_{215\ \mathbf{K}}^{(b)}$ (**) | $h_{235\ \mathbf{K}}^{(b)}$ |
| MD (km) | | −0.47 (1.72) | −0.15 (0.34) | −0.35 (−0.25) | 0.28 | 0.02 | −0.34 | 0.31 (1.37) | −0.03 (0.42) |
| RMSD (km) | | 3.66 (2.28) | 2.38 (1.25) | 1.66 (1.34) | 1.58 | 1.55 | 1.56 | 2.77 (1.90) | 2.57 (1.62) |
| r | | 0.08 (0.72) | 0.45 (0.79) | 0.69 (0.77) | 0.71 | 0.72 | 0.74 | 0.17 (1.69) | 0.52 (0.76) |
| Linear regressor | a | 0.14 (0.73) | 0.64 (0.84) | 0.79 (0.81) | 0.80 | 0.78 | 0.77 | 0.25 (0.68) | 0.84 (1.01) |
| | b (km) | 9.99 (1.39) | 3.09 (1.00) | 1.79 (1.56) | 1.06 | 1.36 | 1.58 | 7.63 (2.07) | 1.25 (−0.53) |
| | RMSD (km) | 3.23 (1.41) | 2.28 (1.17) | 1.57 (1.26) | 1.51 | 1.49 | 1.45 | 2.44 (1.21) | 2.55 (1.57) |
| $h^{(0.9)}$ | a | −0.47 (0.96) | 0.93 (0.94) | 0.93 (0.95) | 0.99 | 0.98 | 0.96 | −0.76 (0.84) | 1.11 (1.11) |
| | b (km) | 21.42 (0.34) | 1.56 (1.33) | 1.95 (1.75) | 0.98 | 1.46 | 1.97 | 22.52 (1.77) | 0.34 (0.20) |
| | MD (km) | 4.74 (1.60) | 0.87 (1.16) | 1.11 (1.17) | 1.21 | 1.35 | 1.39 | 4.18 (1.41) | 1.15 (1.47) |

(*) $h_{210\ K}^{(b)} = h_{trop}^{(b)}$ in the 68% of cases. (**) $h_{215\ K}^{(b)} = h_{trop}^{(b)}$ in the 17% of cases.

Even if we omit the stratospheric altitudes, a correlation coefficient not exceeding 0.74 is a discrete but not an extremely good result. Indeed, the goodness of these estimators lies in another aspect. By looking at the graphs of Figures 12 and 13, with the exception of the $h_{\widetilde{\gamma}_i}^{(tp)}$ cases, we see that the points distribution mainly concentrates around the diagonal axis, it blurs below the diagonal but leaves the area ~1.5 km above the diagonal nearly empty. The same is true for the scatterplots of $h_{230\,\text{K}}^{(RO)}$ and $h_{\varrho,5\times10^{-5}}^{(s,30)}$ with $h_{\varrho,5\times10^{-5}}$ in Figure 11, if we limit ourselves to the tropospheric altitudes. This means that even if the estimators $h_{T_i}^{(RO)}$ and $h_{\varrho,r_i}^{(s,RH)}$ do not completely succeed in accurately predicting the dry air starting altitudes defined in Section 3, probably due to the variability of the relative humidity together with the stratospheric problem, nevertheless, they could be very useful in setting an upper limit to the distribution of $h_{\varrho,r_i}$. Such a property can be highlighted by the altitude $h^{(0.9)}$, defined for each estimator $h_{est}$ as follows: once divided, the estimator range in four intervals is delimited by its quartiles, $h^{(0.9)}$ is a first order polynomial of the estimator that, for each interval, minimizes the square of its local mean deviations from the starting altitudes, exceeding at the same time at least 85% of the starting altitudes. Besides, it is additionally required that it exceeds at least 90% of all the starting altitudes, globally.

More formally, denoted by $h_{est,i}$, $i = 1, \cdots 4$, the subset of estimators that fall in the $i$-th quartile interval, by $h_{dry,i}$ the corresponding subset of the dry air starting altitudes, by $\overline{h}_{est,i}$ and $\overline{h}_{dry,i}$ their respective means, and finally with $q_\alpha(\cdot)$ the $\alpha$-quantile of a distribution, then

$$h^{(0.9)} = a \cdot h_{est} + b,$$

where the real numbers $a$, $b$ minimize the functional

$$F(a,b) = \Theta\left[q_{0.9}\left(h_{dry} - a \cdot h_{est} - b\right)\right] +$$

$$\sum_{i=1}^{4}\left\{\frac{\left(\overline{h}_{dry,i} - a \cdot \overline{h}_{est,i} - b\right)^2}{2} + \Theta\left[q_{0.85}\left(h_{dry,i} - a \cdot h_{est,i} - b\right)\right]\right\}, \tag{17}$$

with

$$\Theta[x] = \begin{cases} 0 & if \quad x \leq 0 \\ +\infty & if \quad x > 0 \end{cases}.$$

For the mathematical method used to derive $a$, $b$, we refer to the Appendix A. The values of $a$, $b$, together with the minimized mean deviation

$$\text{MD} = \overline{h}_{dry} - a \cdot \overline{h}_{est} - b = \frac{1}{4}\sum_{i=1}^{4}\left(\overline{h}_{dry,i} - a \cdot \overline{h}_{est,i} - b\right),$$

are reported in Table 2, with the corresponding $h^{(0.9)}$ straight lines shown in Figures 11–13, together with the quartiles of the estimators.

The idea to consider subintervals is motivated by the request that $h^{(0.9)}$ could work well as the upper bound in every portion of the estimator range. This ensures in particular that, once estimated, $a$, $b$ by the present statistical analysis, $h^{(0.9)}$ could be rightly applied even at those latitudes and longitudes less represented in this study—that is, those places in Figure 9 not covered enough by the events points—whose altitudes may have a different distribution from the global one and may be more concentrated in some subset of the whole dataset.

As part of the correlation coefficients, in order to test the estimation procedures described above, we can compare the uncertainties that we achieve with these estimators—that is, their RMSDs—with the standard deviations of the dry air starting altitude distributions. The last are reported in the first row of Table 2 together with their mean values. In other words, we can check how much better we succeed in applying this estimation procedure with respect to simply using the mean value of the dry air starting altitudes as estimator. With the $h_{T_i}^{(RO)}$ altitudes, we can estimate the quantities $h_{\varrho,10^{-4}}$, $h_{\varrho,1.5\times10^{-4}}$ and $h_{\varrho,2\times10^{-4}}$ with an RMSD that is ~20% lower than the respective standard deviations, a percentage that increases to ~25% if we consider linear regressors; we have a similar result for $h_{\varrho,2.5\times10^{-4}}$ too, with an improvement of ~16% that rises to 22% with the linear regressor. In contrast, when the stratospheric percentage increases, we achieve almost no gain for $h_{\varrho,5\times10^{-5}}$, while for $h_{\varrho,10^{-5}}$ it is practically better to revert to its mean value rather than applying the (best) estimator $h_{210\text{ K}}^{(RO)}$. The same is true for the $h_{\varrho,r_i}^{(s,RH)}$ estimators, even if the results are now slightly worse than the $h_{T_i}^{(RO)}$ ones, in particular when we do not consider linear fittings. However, it is with the lines $h^{(0.9)}$ that we register the best improvements. First, we can see from Table 2 that in all cases, with the exception of $h_{\varrho,10^{-5}}$, $h^{(0.9)}$, the mean deviations are lower than the RMSDs of the estimators; this is true even when the linear fitting is considered, if we further exclude the altitudes $h_{\varrho,2\times10^{-4}}$ and $h_{\varrho,2.5\times10^{-4}}$. However, the mean deviations of $h^{(0.9)}$ and the RMSDs of the estimators or the standard deviations of the starting altitudes have a different statistical meaning. Indeed, by assuming a Gaussian distribution for a dry air starting altitude, we know that 90% of the samples lie within approximatively 1.3 standard deviations above the mean. Using this value as a basis of comparison, that is, if we compare the altitude $h^{(0.9)}$ derived by $h_{T_i}^{(RO)}$ or $h_{\varrho,r_i}^{(s,RH)}$ with the 0.9-quantiles of the starting altitude distributions, after a few calculations we achieve an improvement for the upper bound estimation of the starting altitude that goes from ~33% for $h_{\varrho,2.5\times10^{-4}}$ to ~67% for $h_{\varrho,5\times10^{-5}}$. Besides, $h^{(0.9)}$ performances are not affected by water vapor presence in the stratosphere, except when the fraction of dry air starting altitude percentage greater than the tropopause height exceeds 10%, as in the case of $h_{\varrho,10^{-5}}$.

For the $h_{\widetilde{\gamma}_i}$ estimators, we note from the graphs in Figures 11 and 12 that their point distributions are much more scattered than the $h_{T_i}^{(RO)}$ and $h_{\varrho,r_i}^{(s,RH)}$ ones, resulting in lower correlation coefficients and higher RMSDs. Indeed, by looking at several vertical profiles, we have seen that the appearance of water vapor concentrations is always accompanied with some deviation of bending angle profile with respect to its total variation. Nevertheless, the amount of this deviation is extremely variable, even with respect to the same increment of water vapor concentration. In conclusion, it is very difficult to characterize a specific value of water vapor concentration with a given value of $\widetilde{\gamma}$ and so it is not easy to predict the dry air starting altitudes from the bending angle total variation altitudes.

In Section 6, we will test if the $h_{\widetilde{\gamma}_i}^{(25)}$ altitudes, the only altitude from those defined in Section 4 that can be applied above the tropopause, can at least detect water vapor presence in the stratosphere.

### 5.2. Wet Refractivity Starting Altitudes Estimations

For the wet refractivity starting altitudes, we can apply similar considerations from the previous subsection. By looking at Figure 14, we note that $h_{N1}$ and $h_{N2}$ starting altitudes present a bimodal distribution, with the second modes above 15 km that again have to be attributed to water vapor in the stratosphere, as we deduced from Figures 15 and 16. In particular, in Figure 15 we can see how the $h_{N1}$ altitudes that fall in the stratosphere, exceeding 10% of the total, mainly determine the $h^{(0.9)}$ line. Neglecting the stratospheric altitudes, $h_{N1}$ and $h_{N2}$ show again a gaussian behavior, with even less skewness than the $h_{\varrho, r_i}$ altitudes. Probably due to its definition, $h_{N2}$ has a larger standard deviation than $h_{N1}$, despite having fewer samples in the stratosphere (see Table 2).

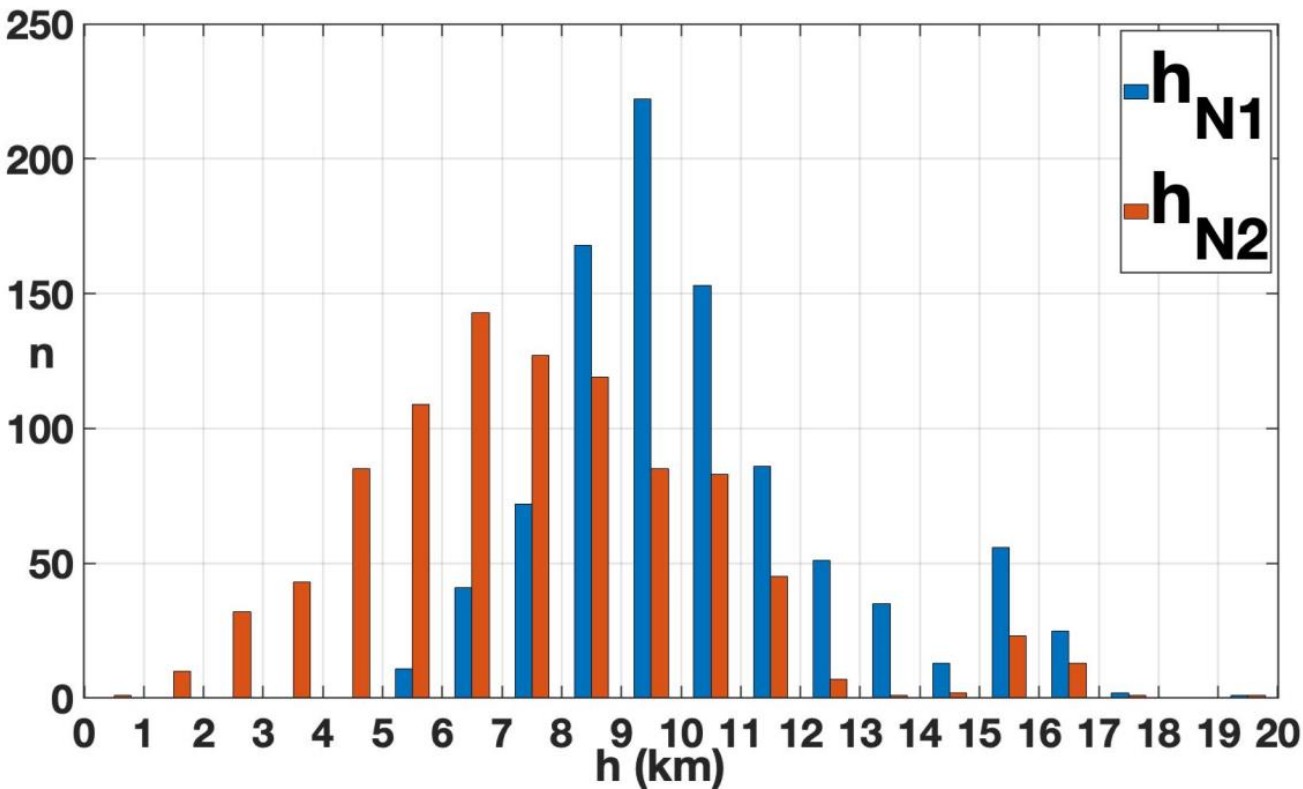

**Figure 14.** Distribution histograms for the dry air starting altitudes $h_{N1}$ and $h_{N2}$. Means and standard deviations for these distributions are reported in Table 2.

From Table 2, we see that the $h_{N1}$ estimation, as in the $h_{\varrho,10^{-5}}$ case, is not well performed by all the methods considered in this study, if compared with the $h_{N1}$ mean value and standard deviations. For $h_{N2}$, it is again best approximated by a temperature-based estimator, $h_{230\ \text{K}}^{(RO)}$, with an RMSD less than ~8% with respect to the $h_{N2}$ standard deviation (~11% if we consider its linear regressor), while for $h_{N2}^{(s,RH)}$, the best estimation is achieved when we set $RH = 40\%$. However, with this higher value of relative humidity, the wet refractivity threshold is never reached in some RO events, so that there are several points along the $y$-axis in the top-right graph of Figure 16, contrary to the case with the estimator $h_{230\ \text{K}}^{(RO)}$, whose scatterplot with $h_{N2}$ is shown in the top-left graph of the same figure. For the $h^{(0.9)}$ lines, we have again an improvement of ~60% with respect to the 0.9 quantile of the $h_{N2}$ distribution. Finally, for the $h_{\widetilde{\gamma}_i}$ altitudes, again they result in the worst estimators.

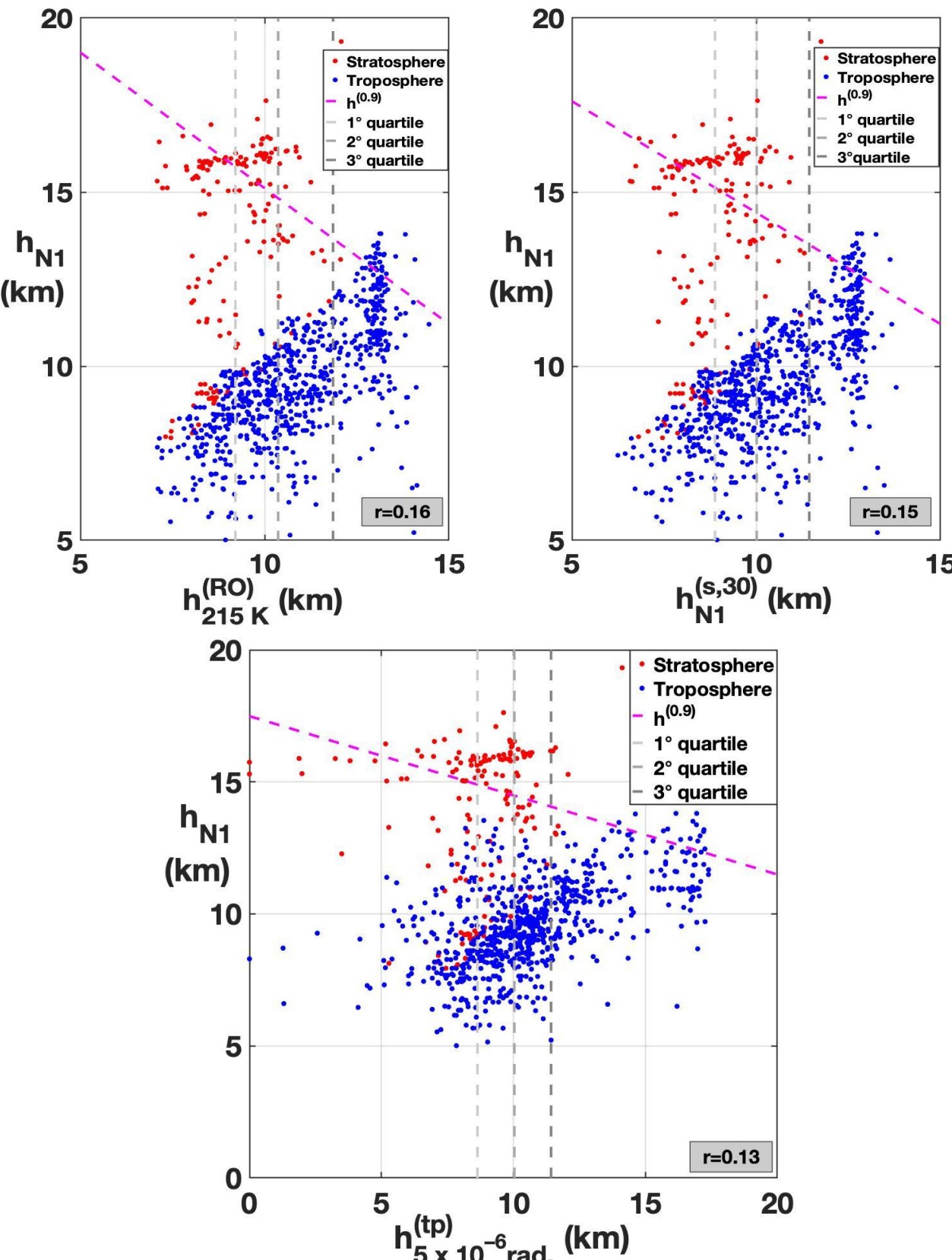

**Figure 15.** Scatterplots among the dry air starting altitudes $h_{N1}$ and the three best estimators $h_{215\,K}^{(RO)}$ (**top left**), $h_{N1}^{(s,30)}$ (**top right**) and $h_{5\times10^{-6}rad}^{(tp)}$ (**bottom**). Correlation coefficients (r) are reported bottom right, for each graph. Red points denote starting altitude falling in the stratosphere, blue points in the troposphere. Purple broken lines denote the $h^{(0.9)}$ straight lines, while grey broken lines are the estimators' quartiles.

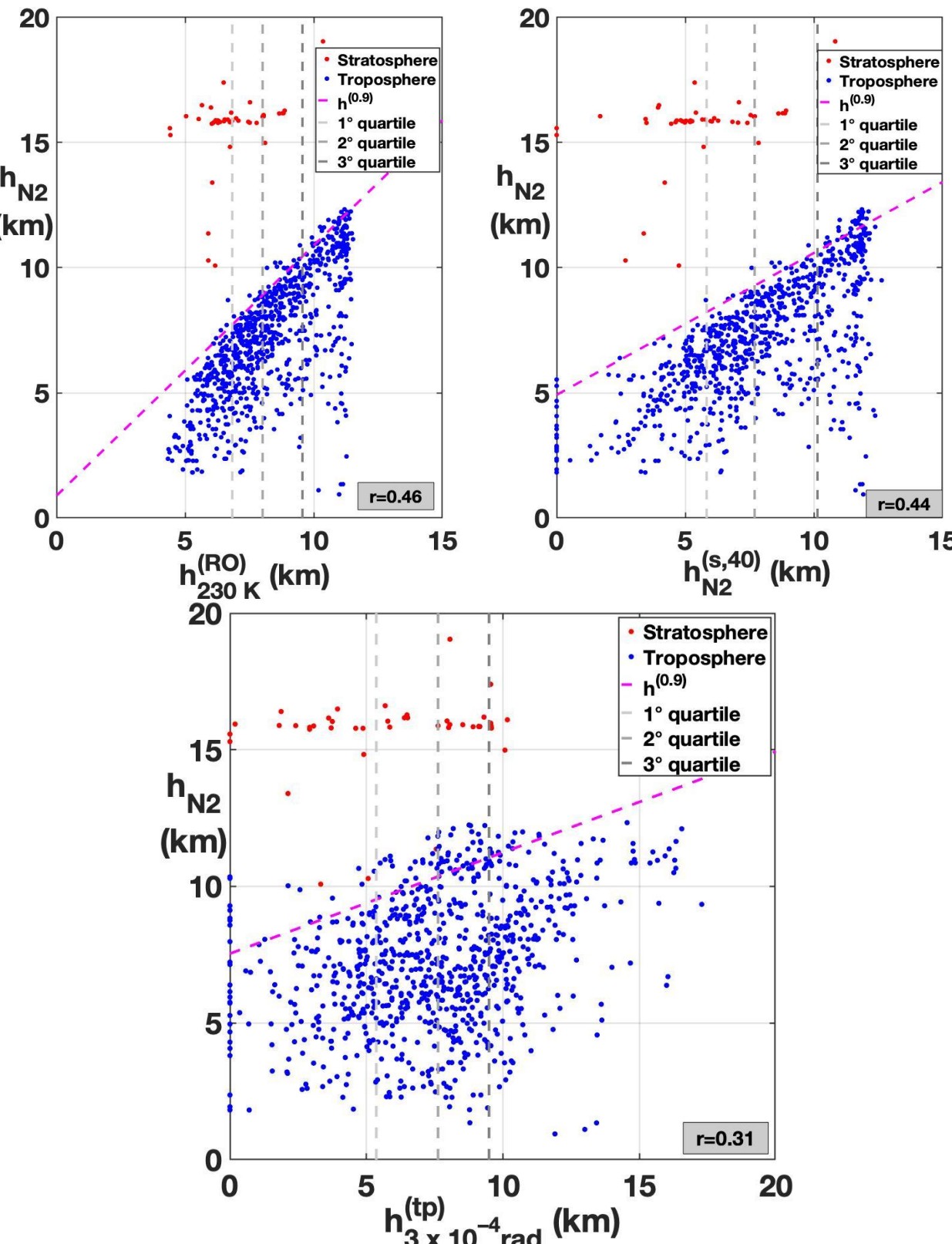

**Figure 16.** Scatterplots among the dry air starting altitudes $h_{N2}$ and the three best estimators $h_{230\,K}^{(RO)}$ (**top left**), $h_{N2}^{(s,40)}$ (**top right**) and $h_{3\times10^{-4}rad}^{(tp)}$ (**bottom**). Correlation coefficients (r) are reported bottom right, for each graph. Red points denote starting altitude falling in the stratosphere, blue points in the troposphere. Purple broken lines denote the $h^{(0.9)}$ straight lines, while grey broken lines are the estimators' quartiles.

## 6. Water Vapor in the Stratosphere

Water vapor concentration in the stratosphere is usually very low, $\varrho_r \sim 10^{-6}$. This is due to the cold tropopause temperature, which causes most of the water vapor coming from the troposphere to condense, forming denser than air agglomerates—water, ice aerosols—unable to reach the higher atmospheric layers. Nevertheless, as we can see from Figures 11 and 12 and from Table 2, in few cases, water vapor reaches non-negligible values ($\varrho_r > 5 \times 10^{-5}$–$10^{-4}$) of concentration in the stratosphere. These higher than usual concentrations are mainly caused by direct injection from an overshooting convention mechanism and methane oxidation [64]. Monitoring water vapor concentrations in the stratosphere is very important in the context of climate change: some studies [65] calculate a stratospheric water vapor impact on global warming of 5–10% from all greenhouse gases, even if others [66] underestimate such an effect.

As said before, with two of our proposed methods to localize threshold values for water vapor concentrations, that is, by temperature and by saturation pressure, we cannot investigate what happens above the tropopause. However, with the irregular bending angle profile method, when we set $h_0 > h_{trop}$ in definition (7), we could, in principle, detect water vapor in the stratosphere. So, in this section we will test the bending angle method limited to the stratosphere, hoping that in this restricted region it could work better than in the general case.

Before exploring such possibilities, we spend some words about the method used to determine the tropopause. In a comparative study such as the present one, this is a delicate task, as it is important to adopt a definition that, when applied both to the RO dataset and to the radiosonde dataset, gives tropopause altitude estimations that are as close as possible. According to the World Meteorological Organization (WMO) [67], the tropopause is defined as "the lowest level at which the (temperature) lapse rate decreases to 2 °C/km, provided that the average lapse rate between this level and all higher levels within 2 km does not exceed 2 °C/km". This Lapse Rate definition for the Tropopause (LRT), introduced in 1957 and still widely used, presents nevertheless some inconveniences: first, for some temperature profiles such a lapse rate condition is not always achieved, or it is achieved at altitudes greater than 30 km. Besides, it is sometimes affected by the vertical resolution of the temperature profile. For this reason, some authors [68] consider a weakened or simplified version of the lapse rate tropopause, which uses only the local lapse rate and the 2 km average. Second, in the tropics, the coldest point of the temperature (CPT) is well marked, and more meaningful than the temperature lapse rate to identify the tropopause. Other possible tropopause definitions concern ozone or other gas concentrations [62,69], potential vorticity [70], and definitions based on the bending angle profile in an RO, corresponding to an irregular pattern [6] or to a deviation from bending angle models with a constant temperature lapse rate hypothesis, as that of Hopfield [71]. In this study, temperature being the only quantity available for both the datasets, we opted for a temperature-based definition for the tropopause.

In Figure 17, we considered four possible variants for this. In Figure 17a, we report the scatterplot of the LRT altitudes with the temperatures given by the radiosonde excursion against the LRT altitudes obtained by the (dry) temperatures from the ROs, while in Figure 17b the same scatterplot with the tropopause defined by the coldest point. In Figure 17c,d we consider two additional tropopause definitions: for Figure 17c, we proposed a weak version of the CPT, defined as the lowest level at which temperature as a local minimum, more precisely a minimum limited to an interval of 2 km centered at the considered level; for Figure 17d, we considered an even more simplified version of the WMO definition than that in [68], by requiring just the mean lapse rate of 2 °C/km in an interval of 2 km centered at the given level without the local condition of lapse rate. For all the adopted definitions, we restricted the tropopause in a vertical window going from 6 km to 12 km at the poles and from 13 km to 21 km at the equator. If the required condition is not satisfied in those intervals, the upper bound interval was adopted for the tropopause

altitude. For the definition of Figure 17d, which we call the weak LRT definition, if the required condition is not satisfied, then the weak CPT was applied.

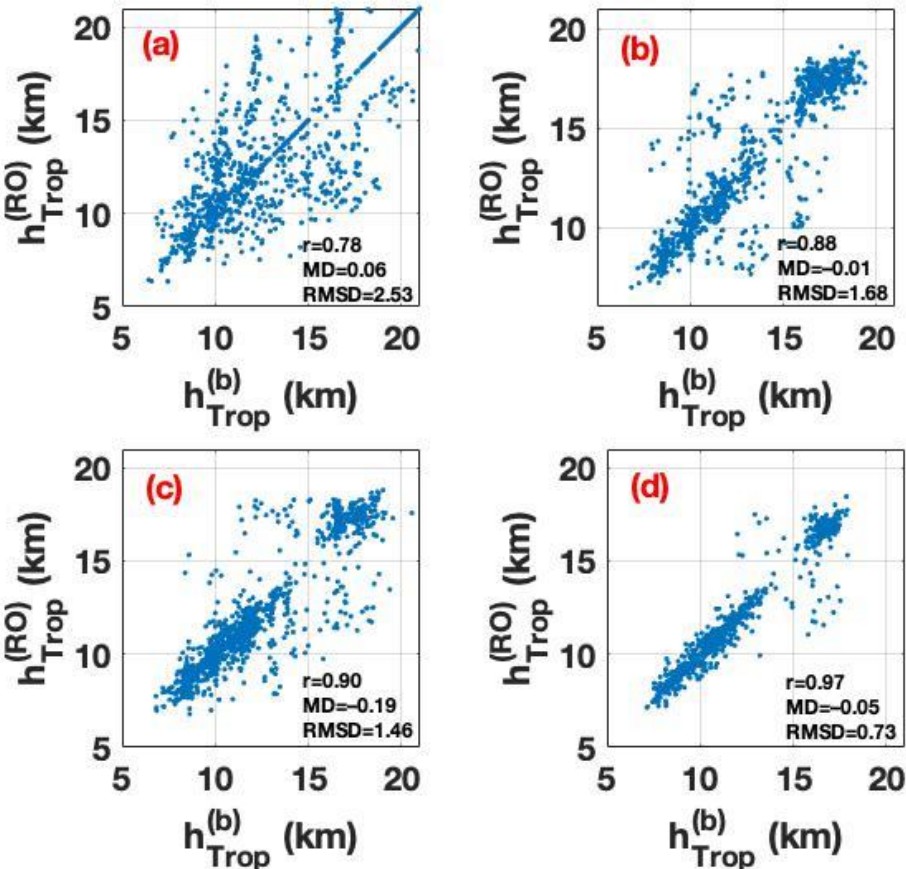

**Figure 17.** Scatterplots of the tropopause altitude as estimated by the real temperature profile (*x*-axis) versus the tropopause altitudes by the RO observations (*y*-axis), for (**a**) the LRT, (**b**) the CPT, (**c**) the weak CPT and (**d**) the weak LRT definitions.

As one can see, the weak LRT definition gives the best results concerning correlation and RMSDs. The points are highly concentrated along the diagonal, with some scattering just around 14 km. The weak CPT also gives good results in the comparative study. Note that, with the exception of the LRT definition, the tropopause distributions for the last two definitions are not very dissimilar with respect to the CPT distribution, for both the balloon and RO profiles, a part of a slight underestimation for the higher tropopause levels. So, we can correctly adopt the weak LRT in our study.

Turning back to water vapor concentrations in the stratosphere, in Figure 18, we plot both the mean mixing ratio vertical profile (blue, identical to that of Figure 3) then the conditioned mean mixing ratio vertical profile (red), the latter by selecting only those profiles for which $h_{\varrho,5\times10^{-5}} > h_{trop}$. The altitudes are scaled with respect to the tropopause height. As one can see, the mixing ratio behavior in the case of $h_{\varrho,5\times10^{-5}} > h_{trop}$ is more complex than the general one: we expect to find dry air starting altitudes above ~1.6 $h_{trop}$, together with a dry air window around the tropopause, where humidity concentration reaches a minimum, while the wet stratosphere is mainly concentrated between $1.2 \div 1.6\ h_{trop}$. Besides, observe that below ~0.3, the tropopause altitude air with stratospheric water vapor is mainly drier, with respect to the general case.

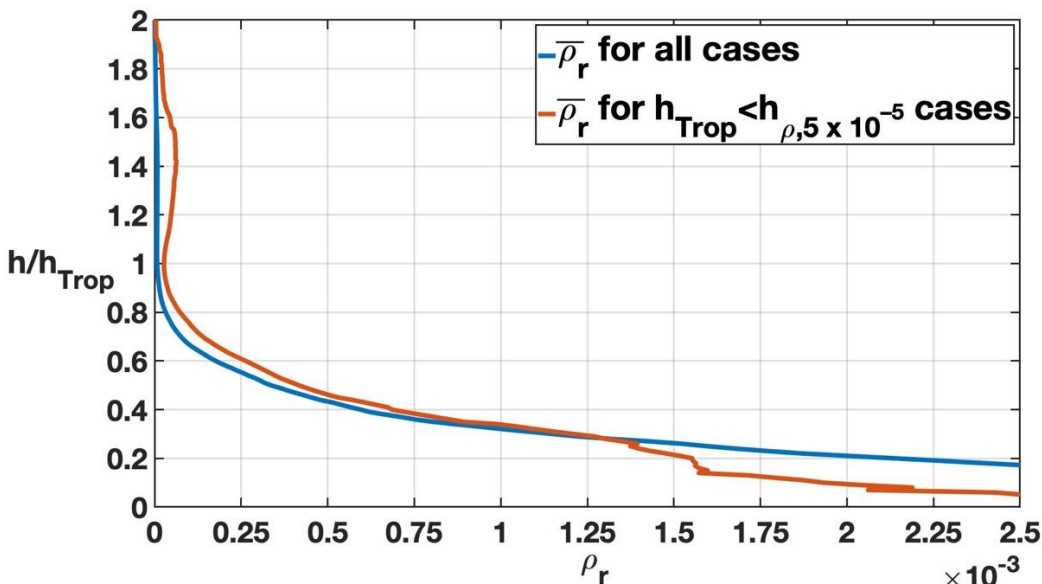

**Figure 18.** Mean values of mixing ratios for all the cases (blue) and for the cases with the starting altitude $h_{\varrho,5\times10^{-5}}$ higher than the tropopause (red). Altitudes are scaled with respect to tropopause elevations.

In this more complex situation, we limit ourselves to simply detecting the presence of water vapor in the stratosphere, instead of determining dry air threshold altitudes. We try to do this by comparing the probabilities for dry air starting altitudes to lie above the tropopause with the probabilities for the same event conditioned this time by the estimators to be in the stratosphere as well. If these probabilities are very different, than we can find a dependency between the conditioned and the conditioning event, that is we can make predictions about water vapor in the stratosphere from RO observations. In Table 4, we report the results obtained following this approach. For this study we just considered the starting altitudes $h_{\varrho,10^{-5}}$, $h_{\varrho,5\times10^{-5}}$, $h_{\varrho,10^{-4}}$, $h_{N1}$ and $h_{N2}$, for which the probability of falling in the stratosphere, reported in the first row, was not extremely low or null, while as estimators we used the altitudes $h_{\gamma_i}^{(25)}$. In the remaining rows, the probabilities for the starting altitudes to lie in the stratosphere, conditioned by the estimator to be above the tropopause as well, are reported. All the probabilities are given in percentages.

**Table 4.** Probabilities for the dry air starting altitude to be in the stratosphere (first row) and the probabilities for the same events conditioned by the estimators $h_{\gamma_i}^{(25)}$ to be in the stratosphere as well.

| | | $h_{\varrho,10^{-5}}$ | $h_{\varrho,5\times10^{-5}}$ | $h_{\varrho,10^{-4}}$ | $h_{N1}$ | $h_{N2}$ |
|---|---|---|---|---|---|---|
| Stratospheric probability (%) | | 34.76 | 5.37 | 0.99 | 20.29 | 4.71 |
| Stratospheric conditioned probability with respect to the estimator to be above the tropopause (%) | $h_{5\times10^{-6}rad}^{(25)}$ | 39.30 | 6.15 | 1.12 | 22.80 | 5.31 |
| | $h_{10^{-5}rad}^{(25)}$ | 39.32 | 6.22 | 1.13 | 22.63 | 5.37 |
| | $h_{5\times10^{-5}rad}^{(25)}$ | 38.03 | 5.61 | 1.21 | 21.67 | 5.00 |
| | $h_{10^{-4}rad}^{(25)}$ | 37.54 | 4.98 | 1.33 | 21.10 | 4.65 |
| | $h_{1.5\times10^{-4}rad}^{(25)}$ | 37.99 | 5.02 | 1.43 | 20.61 | 4.84 |
| | $h_{2\times10^{-4}rad}^{(25)}$ | 38.30 | 4.84 | 1.55 | 20.50 | 4.64 |

As we can see, the conditioned probabilities are just slightly different from the original ones; the bending angle altitude estimators in the stratosphere constitute almost an independent event with respect to the starting altitudes in the stratosphere. So, even this RO method fails to detect water vapor in the stratosphere.

### 7. Conclusions

Accuracy, global coverage and continuous operativity due to its insensitivity to clouds make the GNSS-RO a great instrument for weather forecasts and climate change monitoring. The flipside is that from RO observations alone we cannot predict water vapor concentrations, and RO estimations of atmospheric temperature and pressure actually lose accuracy in humid air, as we have seen in Figure 3 and Table 1. While this problem is usually solved by considering additional information, in the form of data from other experimental sources or background data such as forecasts, nevertheless in some stand-alone methods, such as BPV, it is important to determine, by means of the same RO data, the threshold altitudes above which RO techniques keep their accuracy in estimating the main atmospheric parameters. In the present study, after determining the dry air starting altitudes by water vapor mixing ratios and wet refractivity under the criterion to preserve RO accuracy, we considered three possible methods to estimate these altitudes: by (dry) air temperatures, by water vapor saturation pressure, and by bending angle profile irregularity. For the first two estimators, they showed similar results, even if the air temperature predictions were slightly better. They cannot be applied to the stratospheric region, so they lack in accuracy in estimating $h_{\varrho,10^{-5}}$ and $h_{N1}$ altitudes, which frequently fall above the tropopause. However, for the other starting altitudes, the estimations were good enough, if compared with the information that we can achieve from the starting altitude distributions themselves: in particular, for the *safe* condition of over-estimating the starting altitudes with 90% of probability, with the $h^{(0.9)}$ line we can optimize such a condition up to reduce the mean distance from the starting altitudes by more than half the same distance for the 90 percentile of the starting altitude distributions themselves.

For the irregular bending angle pattern estimator, via the deviation of the bending angle profile with its total variation, we did not achieve results as good as the other two methods. Indeed, even by removing the stratospheric altitudes, the linear correlation coefficients assumed very low values, not exceeding 0.55. We also did not achieve good results in detecting water vapor in the stratosphere with this method.

**Author Contributions:** All authors listed have made a substantial, direct and intellectual contribution to the work, and approved it for publication. All authors have read and agreed to the published version of the manuscript.

**Funding:** This research was carried out in the framework of the project Advanced EO Technologies for studying Climate Change impacts on the environment—OT4CLIMA which was funded by the Italian Ministry of Education, University and Research (D.D.2261 del 6.9.2018, PON R&I 2014–2020 e FSC).

**Institutional Review Board Statement:** Not applicable.

**Informed Consent Statement:** Not applicable.

**Data Availability Statement:** The raw data supporting the conclusions of this article will be made available by the authors, without undue reservation.

**Conflicts of Interest:** The authors declare no conflict of interest.

### Appendix A

In a similar way as performed in [37], we approximate the functional $F(a,b)$ of Equation (17) with the following functional

$$\widetilde{F}(a,b) = \lambda^2 \exp\left[\frac{q_{0.9}\left(h_{dry} - a \cdot h_{est} - b\right)}{\lambda}\right] + \sum_{i=1}^{4}\left\{\frac{\left(\overline{h}_{dry,i} - a\cdot\overline{h}_{est,i} - b\right)^2}{2} + \lambda^2 \exp\left[\frac{q_{0.85}\left(h_{dry,i} - a \cdot h_{est,i} - b\right)}{\lambda}\right]\right\},$$

where the parameter $\lambda$ is a length and assumes very low values with respect to the considered altitudes. Indeed, as $\lambda \to 0^+$, we have the following punctual convergence:

$$\lambda^2 \exp\left[\frac{x}{\lambda}\right] \to \Theta[x] = \left\{ \begin{array}{ll} 0 & if \quad x \leq 0 \\ +\infty & if \quad x > 0 \end{array} \right. .$$

Once a value was fixed for $\lambda$ (in our case, $\lambda = 10^{-3}$ km), we applied the steepest descent method in order to minimize $\widetilde{F}$. Actually, admitting $\widetilde{F}$ several local minima, we opted for considering numerous fixed values for the parameter $a$, uniformly distributed in the interval $-0.7 \leq a \leq 2$, while the steepest descent method was applied just with respect to the parameter $b$. Denoted by $b_a$ the minimizing value of $b$ for a given fixed value of $a$, we selected the best couple $(a, b_a)$ by a direct comparison among the various values of $\widetilde{F}(a, b_a)$ so obtained.

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
