# Peer review of "Comparisons of Different Methods to Determine Starting Altitudes for Dry Air Atmosphere by GNSS-RO Data"

_atmosphere, doi:10.3390/atmos12101276_

Round 1

Reviewer 1 Report

This manuscript presents a very comprehensive study of dry temperature staring altitude using multiple methods in the comparison. The study was designed and implemented in a logical and clear way. The manuscript is very informative and will benefit any radio occultation neutral temperature products users. There are some issues I think the authors should address, they are mostly minor. The water vapor in the atmosphere is largely location- and seasonal- dependent, especially latitudinal-dependent. The abundance of moisture is greater in equatorial areas than in high-latitude or polar areas. If the overall averaging would smooth out some variations due to different moisture? This also applies to the selected radio occultation-radiosonde comparison that only contains data from January/February and May. When multiple mixing ratios are considered with respect to the temperature deviations, is it necessary also to consider the uncertainty of the radio occultation measurements?

There are plenty of language wording errors including missing citations, wrong symbols, spaces, and wrong figures numbers. They must be corrected before the manuscript can be published. And, some figures with multiple sub-figures need to be organized in a better format such as alignment and axis limit.

Author Response

About the question concerning water vapor variability, our aim was to find some parameters by which we can estimate the dry air starting altitudes, parameters that possibly take into account of such variations better than the average values. For this reason, in the article we compare RMSDs of the estimators with the standard deviation of the dry air starting altitude distribution.

About the dates of the RO events, we unfortunately don’t dispose of additional events in other days. Nevertheless, since the events are localized both in the northern and southern emisphere, we (partly) cover the season variability.  

The uncertainty in radio occultation measurements are implicitly considered for what concerns mixing ratios. We considered deviations of dry temperture from that achieved by the radiosonde, in the dry part of the atmosphere, as an estimation of the RO uncertainty.  

A lot of misprints were corrected.

Reviewer 2 Report

The article is interesting, it touches many important threads, but above all it is very chaotic and it is completely unclear what comes from what.

  1. The abstract does not show what is in the article. Moreover, it looks like a copied excerpt from another article. There are references to formulas, coefficients, but apparently the symbols were not copied and are replaced with commas. This is the case in many places in the article, for example on pages 1 (37, 41), 2 (56), 3 (120). This introduces a lot of confusion and changes the meaning of the content.
  2. There is no literature review at all, which is unacceptable in such articles. Moreover, the authors do not cite bibliographic entries from the bibliography.
  3. The authors in the introduction do not introduce the topic. They start methodological considerations right away. This makes it difficult to understand what the article is actually about. The authors also did not specify what they wanted to achieve by testing the three methods. So it is difficult to verify whether any goal has been achieved.
  4. The captions of figures and tables are definitely too long. They should fit in one short title, and the rest of the content should be included in the article.
  5. The numbering of the figures is completely wrong. Please correct, because in the middle of the article the numbering starts with "3" again. Earlier, reference was made to the results presented in Fig. 13, which is not present at all. This would also indicate a copy of other content not sourced from this article.

Author Response

  1. Abstract was changed and we think the article is better introduced now.
  2. Actually, we didn’t fully understand this point. To our knowledge, there are no other articles about dry air starting altitudes except that of Kursinski [4]. If this is not true, we would appreciate if you can suggest some of them. About the citations number, they are now added.
  3. We changed the introduction, and now the aims of the article are better described. We added a new section for the methodology part that was previously in the introduction.
  4. We reduced captions for some figures. About table 2 and 3, we choose to leave them unchanged because such tables cointain a lot of data, and we think that a fully description in the caption could be benefit for the reader, rather then finding it in the text corpse.
  5. We correct figure numbering. A lacking figure (Figure 18) was added.

Round 2

Reviewer 1 Report

The major review comments are addressed well, added and adjusted text contents in the manuscript are good. It is good to be considered for publication after some minor editorial errors are corrected, including missing citations, wrong spaces (probably related to missing citations), a blank page (#24). 

Reviewer 2 Report

The article has a high substantive value. It presents a new approach to the subject. Still, it is quite chaotic. Moreover, it seems that some of the publication's citations and some mathematical formulas were not readable in the PDF, which makes it difficult to understand the work. There is still no classic literature review.